# Natural external plastron mold of the Triassic turtle *Proterochersis*: An unusual mode of preservation

**Tomasz Szczygielski**[1]*, **Lorenzo Marchetti**[2], **Dawid Dróżdż**[3]

**1** Institute of Paleobiology, Polish Academy of Sciences, Warsaw, Poland, **2** Museum für Naturkunde, Leibniz-Institut für Evolutions-und Biodiversitätsforschung, Berlin, Germany, **3** Nalecz Institute of Biocybernetics and Biomedical Engineering, Polish Academy of Sciences, Warsaw, Poland

* t.szczygielski@twarda.pan.pl

**Data Availability Statement:** All relevant data are within the manuscript. The specimens described belong to their respective collections held at scientific institutions and museums (see

## Abstract

Impressions of vertebrate bodies or their parts, such as trace fossils and natural molds of bones, are a valuable source of information about ancient faunas which may supplement the standard fossil record based on skeletal elements. Whereas trace fossils of animal activity are relatively common and actively studied within the field of ichnology, and natural impressions of internal or external surfaces are a frequent preservation mode in fossil invertebrates, natural molds of bones are comparatively rare and less extensively documented and discussed. Among them, internal molds (steinkerns) of turtle shells are a relatively well-known form of preservation, but the mechanisms and taphonomic prerequisites leading to their formation are poorly studied. External shell molds are even less represented in the literature. Herein, we describe a historic specimen of a natural external turtle plastron mold from the Triassic (Norian) Löwenstein Formation of Germany–a formation which also yielded a number of turtle steinkerns. The specimen is significant not only because it represents an unusual form of preservation, but also due to its remarkably large size and the presence of a potential shell pathology. Although it was initially interpreted as *Proterochersis* sp., the recent progress in the knowledge of proterochersid turtles leading to an increase in the number of known taxa within that group allows us to verify that assessment. We confirm that the specimen is morphologically consistent with the genus and tentatively identify it as *Proterochersis robusta*, the only representative of that genus from the Löwenstein Formation. We note, however, that its size exceeds the size observed thus far in *Proterochersis robusta* and fits within the range of *Proterochersis porebensis* from the Grabowa Formation of Poland. The marks interpreted as shell pathology are morphologically consistent with *Karethraichnus lakkos*–an ichnotaxon interpreted as a trace of ectoparasites, such as leeches. This may support the previously proposed interpretation of *Proterochersis* spp. as a semi-aquatic turtle. Moreover, if the identification is correct, the specimen may represent a very rare case of a negative preservation of a named ichnotaxon. Finally, we discuss the taphonomy of the Löwenstein Formation turtles in comparison with other Triassic turtle-yielding formations which show no potential for the preservation of internal or external shell molds and propose a taphonomic model for the formation of such fossils.

Institutional abbreviations) and are available for study.

**Funding:** The study was supported by the National Science Centre, Poland (Narodowe Centrum Nauki, https://www.ncn.gov.pl/en) grant no. 2020/39/B/NZ8/01074 awarded to T. Sz. The funders had no role in study design, data collection and analysis, decision to publish, or preparation of the manuscript.

**Competing interests:** The authors have declared that no competing interests exist.

## Introduction

Due to their durable shell and frequent occurrence in faunal assemblages since the Late Triassic, turtles are relatively common in the fossil record and are usually represented by complete shells or shell fragments. In rare cases, if the taphonomic setting promotes such processes, the remains may be preserved through unusual fossilization pathways, e.g., leading to opalization [1] or fossilization of osteocytes and blood vessels [2–4], surrounding soft tissues [5–7], or epidermal scutes [7–9]. When the diagenetic conditions are unfavorable or the specimen is exposed to harsh environment for an extended time before it is collected, the fossilizing or fossilized bone may be subjected to dissolution or severe weathering, making it brittle and subject to destruction during recovery or preparation, or solely leaving behind an internal mold of lithified rock matrix [10]. This mode of preservation is relatively common in the Upper Triassic of Germany, and the natural internal molds of turtle shells, called steinkerns (stone cores), in some cases with remaining parts of the shells, constitute a significant part of the German Triassic turtle collections [11–18]. Depending on the grain size and hardness of the rock matrix, both absolute and relative to bone, such molds in some cases can preserve superior surface detail. In contrast, external molds of the shells, especially in localities and formations with a good fossiliferous potential, are much rarer and represented by only two specimens of Triassic turtles: *Proterochersis robusta* Fraas, 1913 [13] SMNS 16442 (Fig 1), consisting of a partial external mold of the shell, partial steinkern, and several fragments of the actual shell [15, 16, 19, 20], and SMNS 15479 (Fig 2)–thus far undescribed, largely complete imprint of an unusually large plastron with no associated bone remains or steinkern. The aim of this work is to describe and identify the latter specimen, and to interpret its geological context.

## Institutional abbreviations

CSMM, Carl-Schweizer-Museum, Murrhardt, Germany; IVPP, Institute of Vertebrate Paleontology and Paleoanthropology, Chinese Academy of Sciences, Beijing, China; MB, Museum

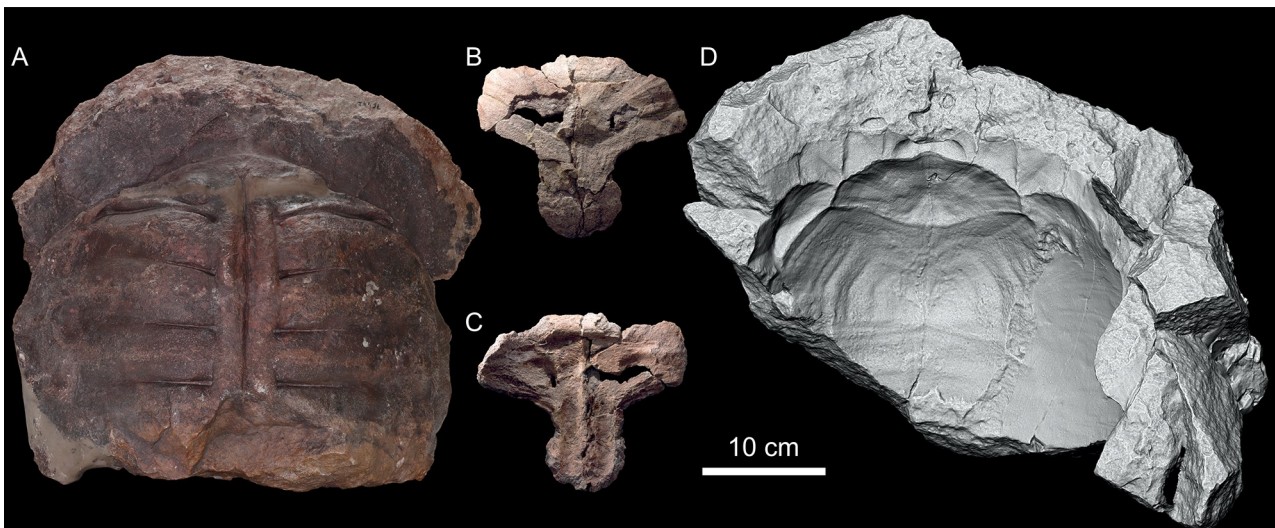

**Fig 1. *Proterochersis robusta* SMNS 16442, an example of three forms of shell preservation within a single specimen. A**, steinkern (natural internal mold) of the shell in dorsal view, with imprints of the visceral surface of the carapace (including ribs) and a faint impression of the carapace periphery. **B**, **C**, nuchal fragment of the carapace (actually preserved bone) in (B) external and (C) visceral view. **D**, partial natural external mold of the carapace (3D model in orthographic view with the Radiance Scaling (Lambertian) shader enabled for increased clarity) with clear imprints of scute borders (sulci) and surface. Note that in this particular case the detail preservation and completeness of the molds is superior to that on the actual shell fragment.

für Naturkunde Berlin, Berlin, Germany; NHMD, Natural History Museum of Denmark, Copenhagen, Denmark; PULR, Universidad Nacional de La Rioja, La Rioja, Argentina; SMNS, Staatliches Museum für Naturkunde, Stuttgart, Germany; SM (former TF), Sirinndhorn Museum, Department of Mineral Resources, Kalasin, Thailand; ZPAL, Institute of Paleobiology, Polish Academy of Sciences, Warsaw, Poland.

## Material and methods

SMNS 15479 is a slab of coarse-grained sandstone (consisting of three parts connected with plaster) from the Löwenstein Formation (former Stubensandstein, Norian, Late Triassic) bearing on its surface a natural imprint of a large turtle plastron (Fig 2). The specimen was found in 1926 in Reichenbach at the river Fils (east of Stuttgart, Germany) in the quarry run by W. Fischer, who gifted it to SMNS. The original specimen is accompanied by a plaster cast, representing the positive of the imprinted plastron, although partially restored and extending beyond what is preserved in the actual fossil.

According to the specimen labels, it was initially referred to the genus *Proganochelys* Baur, 1887 [11] (not published) and subsequently to *Proterochersis* Fraas, 1913 [13]. The first mention of the specimen as *Proterochersis* was made by Wurster [21]. Later it was listed, also without specific attribution, by Broin [20], who provided the measurement of the length of the plastron (47 cm), based on which she estimated the length of the carapace at 50 cm. The only published photograph of SMNS 15479 was presented by Gaffney [18] (fig 68, captioned "*Proterochersis robusta*, SMNS 15479. External mold of plastron") who, however, made no reference to it anywhere in the text. Szczygielski et al. [15] and Szczygielski and Sulej [16] considered the specimen not diagnostic beyond Proterochersidae indet.

SMNS 16442 (Fig 1) is a composite specimen consisting of a steinkern of the anterior and middle part of the shell (Fig 1A), a partial external mold of the anterior portion of the carapace (Fig 1D), a minor fragment of an external mold of the anterior part of the plastron (area of the humeropleural sulcus), fragment of the anterior lobe of the plastron (entoplastron and partial hyoplastra, adjacent to but non-overlapping with the external mold), a part of the anterior (Fig 1B and 1C) and anteroposterior left part of the carapace (just anterior to the bridge), a partial posterior lobe of the plastron (posterior part of both femoral, both complete anal, and fragmentary left caudal scute areas), and several plaster molds and casts. It was found in the middle Löwenstein Formation of Murrhardt, Germany in 1929. The specimen was first mentioned and partly figured by Oertle [19]. Later it was partly figured as *Proterochersis* sp. by Broin [20], mentioned as *Proterochersis robusta* by Gaffney [18]. Karl and Tichy [22] referred it to their newly erected taxon, "*Murrhardtia staeschei*" Karl and Tichy, 2000 [22], which was subsequently synonymized with *Proterochersis robusta* by Szczygielski and Sulej [14], who also provided a short description of SMNS 16442. Eventually, it was described in detail as *Proterochersis robusta* and figured by Szczygielski *et al.* [15] and Szczygielski and Sulej [16].

In addition to the above specimens, the following material of Triassic stem turtle plastra was studied first hand by the authors: *Keuperotesta limendorsa* Szczygielski and Sulej, 2016 [14] SMNS 17757 [14, 23, 24]; *Odontochelys semitestacea* Li et al., 2008 [25] IVPP V 13240, IVPP V 15639, IVPP V 15653 [25–30]; *Palaeochersis talampayensis* Rougier et al., 1995 [31] PULR 068 [31–34]; *Proganochelys quenstedtii* Baur, 1887 [11] MB.1910.45.2, SMNS 16980, SMNS 17203, SMNS 17204 [18, 20, 35–37]; '*Proganochelys*' *ruchae* Broin, 1984 [20] SM2015-1-001 (former TF 1440–6) and SM2017-1-129 (former TF 1440–11) [20]; numerous specimens of *Proterochersis porebensis* Szczygielski and Sulej, 2016 [14], the most relevant being ZPAL V. 39/34, ZPAL V. 39/48, ZPAL V. 39/49, ZPAL V. 39/379; ZPAL V. 39/385, ZPAL V. 39/387 [14–16, 24, 38]; *Proterochersis robusta* CSMM uncat., SMNS 11396, SMNS 12777, SMNS

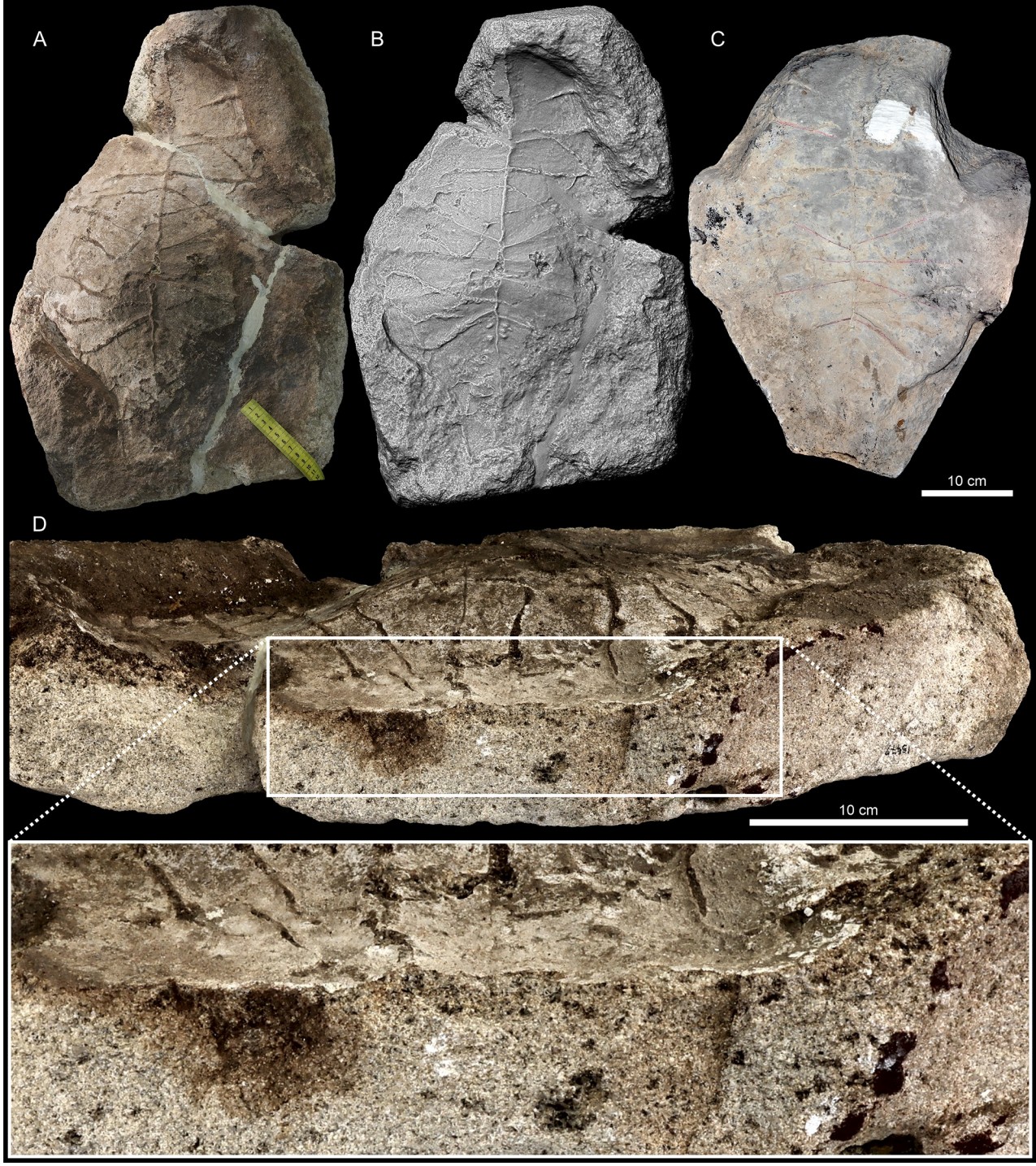

**Fig 2. *Proterochersis robusta* SMNS 15479. A**, photograph and **B**, 3D model of the slab in orthographic view with the Radiance Scaling (Lambertian) shader enabled. **C.** Photograph of the plaster cast, note that some parts are restored. **D.** Photograph of the slab in left lateral view with enlarged natural cross section through the plastron imprint, showing the light gray, finer sediment lining of the plastron imprint and heterogenous discoloration of the matrix.

16603, SMNS 17561, SMNS 17755, SMNS 18440, SMNS 50917, SMNS 56606, SMNS 81917 [13–16, 20, 22]; and Greenland proganochelyid NHMD 190349 and other specimens from the Fleming Fjord Formation in the NHMD collection [39–41].

SMNS 15479, *Proterochersis robusta* SMNS 17561, the carapace of *Keuperotesta limendorsa* SMNS 17757, *Palaeochersis talampayensis* PULR 068, *Odontochelys semitestacea* IVPP V 13240, and *Proganochelys quenstedtii* SMNS 16980 and MB.1910.45.2 were digitized through photogrammetry using Agisoft Metashape Professional 2.0.1 (photographs aligned on the High setting, mesh produced from depthmaps on the Ultra High setting) and automatically scaled based on printed scale-markers.

The plastron of *Proterochersis porebensis* ZPAL V. 39/49, the nuchal part of the carapace of *Proterochersis robusta* SMNS 16642, and the plastron of *Keuperotesta limendorsa* SMNS 17757 were scanned using a Shining 3D EinScan Pro 2X 3D surface scanner with EXScan Pro 3.7.0.3 software, fixed on a tripod with EinScan Pro 2X Color Pack (texture scans), either using an Ein-Turntable (alignment based on features; SMNS 16642; SMNS 17757) or without a turntable, rotating the specimen manually (ZPAL V. 39/49). The number of turntable steps was varied, chosen depending on the fragment. The models were meshed using the Watertight Model and High Detail presets.

To better visualize the morphology of the plastron of SMNS 15479, a virtual positive was made by inverting the model of the original (natural mold, i.e., negative) in MeshLab 2022.02 [42] using the Invert Faces Orientation tool (menu Filters -> Normals, Curvatures and Orientation). Snapshots of the resulting mesh were captured using the Save snapshot tool in orthographic view and with the Radiance Scaling (Lambertian, Lit Sphere, and Grey Descriptor) shader enabled [43]. Additionally, an elevation map was generated in ParaView 5.11.1 [44, 45]. Snapshots of the remaining specimens were captured in orthographic view with the Radiance Scaling (Lambertian or Lit Sphere) shader enabled.

## Geological setting and taphonomy

Remains of Triassic turtles are known from several geological formations and usually found in mudstones, sandstones, conglomerates, or marlstones (Table 1). The most productive, in terms of turtle specimen number, are the Löwenstein Formation of Germany [11–18, 22–24, 46–56], the Grabowa Formation of Poland [14–16, 24, 38, 52, 57], and the Fleming Fjord Formation of Greenland [39–41]. Out of those, only the Löwenstein Formation of Germany has yielded documented internal and external molds of turtle shells, including SMNS 15479. Turtle specimens from Greenland are mostly unpublished, aside from short notes, and will be tackled elsewhere, but no external or internal shell molds are preserved among the collected material.

### Löwenstein Formation

Natural internal or external molds of turtle shells from the Löwenstein Formation include SMNS 15479 (Fig 2), as well as the holotype (SMNS 12777, Fig 3A) and several referred specimens of *Proterochersis robusta* (CSMM uncat., Fig 3D; SMNS 11396, Fig 3B; SMNS 16442, Fig 1; SMNS 16603, Fig 4A, 4B; SMNS 16668; SMNS 17756, Fig 4D; SMNS 17930, Fig 4C; SMNS 19103) and the holotype (GPIT-PV-30000, Fig 5A) and referred specimen (SMNS 10012, Fig 5B) of *Proganochelys quenstedtii* [11, 12, 17, 18, 50, 51].

The most informative fossils of *Proterochersis robusta* are considered to originate from the lower unit of the Löwenstein Formation [14, 22, 46–49], many of them from the locality of Murrhardt. German finds of *Proganochelys quenstedtii* are apparently restricted to the Middle and Upper Löwenstein Formation and the overlying Trossingen Formation [18, 47, 49], the latter not producing internal or external molds of turtle shells. Sadly, not every historic

**Table 1. Triassic turtle-bearing formations and the occurrence of internal and external molds of shells.** The depositional environment provided for turtle-yielding strata only.

| Formation | Matrix | Turtle taxa | Location | Molds | Depositional environment | References |
|---|---|---|---|---|---|---|
| Bull Canyon | Mudstone, siltstone, sandstone | *Chinlechelys tenertesta* | USA | - | Floodplains | [66–69] |
| Fleming Fjord | Mudstone, siltstone, sandstone | 'cf. *Proganochelys*', Testudinata indet. | Greenland | - | Lacustrine | [39–41] |
| Grabowa | Mudstone, claystone, sandstone, conglomerate | *Proterochersis porebensis*, *Proterochersis* cf. *porebensis* | Poland | - | Braided or anastomosed river (Poręba); caliche-like lithification of fluvial channel infillings (Kocury) | [14–16, 24, 38, 52, 53, 57, 63–65] |
| Huai Hin Lat | Shale, limestone, sandstone, conglomerate | '*Proganochelys*' *ruchae* | Thailand | - | Fluvial and lacustrine, fault bounded basins | [20, 70–72] |
| Klettgau (Gruhalde Member) | Marl | *Proganochelys quenstedtii* | Switzerland | - | Playa with fluvial channels | [73, 74] |
| Los Colorados | Mudstone, sandstone | *Palaeochersis talampayensis* | Argentina | - | Floodplains or fluvial with permanent water bodies, or ephemeral fluvial with eolian sand bodies | [31–34] |
| Löwenstein (Stubensandstein) | Shale, claystone, limestone, sandstone | *Keuperotesta limendorsa*, *Proganochelys quenstedtii*, *Proterochersis robusta* | Germany | + | Terminal alluvial plain | [11, 12, 17, 18, 35–37, 50, 51, 75–77] |
| Quebrada del Barro | Claystone, sandstone, conglomerate | *Waluchelys cavitesta* | Argentina | - | Floodplains | [33, 34, 78] |
| Trossingen (Obere Mühle) | Marl | *Proganochelys quenstedtii* | Germany | - | Episodically flooded dry mudflat playa | [18, 37, 75, 76, 79, 80] |

specimen from the Löwenstein Formation has associated precise stratigraphic information beyond "Stubensandstein", but it is currently assumed, based on occurrences in clearly different stratigraphic intervals, that there was no temporal overlap between *Proterochersis robusta* and *Proganochelys quenstedtii* in Germany [14, 47, 49].

The Löwenstein (Stubensandstein) Formation was characterized as representing semi-arid or semi-humid terminal alluvial plains and the dominating lithology is sandstone (clayey, calcareous, or quarzitic) interdigitating with claystone, depending on the cyclically changing water level at the time of deposition [18, 49, 58–61]. Locally, the matrix contains larger clasts of irregular shape. The sandstone rock is hard, predominantly gray, yellowish, or reddish, with individual grain or clast color slightly differing (usually darker) from the color of the surrounding cement, albeit larger color differences are locally observed.

Turtle bones found in the Löwenstein Formation usually are light gray to cream-colored without (*Proganochelys quenstedtii* SMNS 10012, SMNS 15759; *Proterochersis robusta* SMNS 11396, SMNS 16668) or with (*Proterochersis robusta* NHMUK 38650, SMNS 16442, SMNS 16603, SMNS 17561, SMNS 17755, SMNS 17755a, SMNS 17756, SMNS 18440, SMNS 19103, SMNS 56606, SMNS 81917) a reddish tint and infillings of intertrabecular spaces, to various intensities and shades of brown (minute bone remains in *Proganochelys quenstedtii* GPIT-PV-30000; *Proterochersis robusta* SMNS 12777, SMS 17930, SMNS 50917). In one instance (CMSMM uncat.), the bone surface locally attains a bluish shade, even though the bone is reddish-brown in the cross section (Fig 3D). The bones are hard and overall characterized by good surface preservation but, depending on the preparation technique used and weathering, surfaces of some specimens may suffer various degrees of damage.

When strongly weathered or improperly prepared, the bone may be partially or completely destroyed, leaving behind a natural mold of its internal (steinkern) or, less frequently, external surface. Bone fragments with jagged edges, frequently flakes of the bone cortex, may remain attached to the imprint. The holotype *Proganochelys quenstedtii* steinkern presumably from

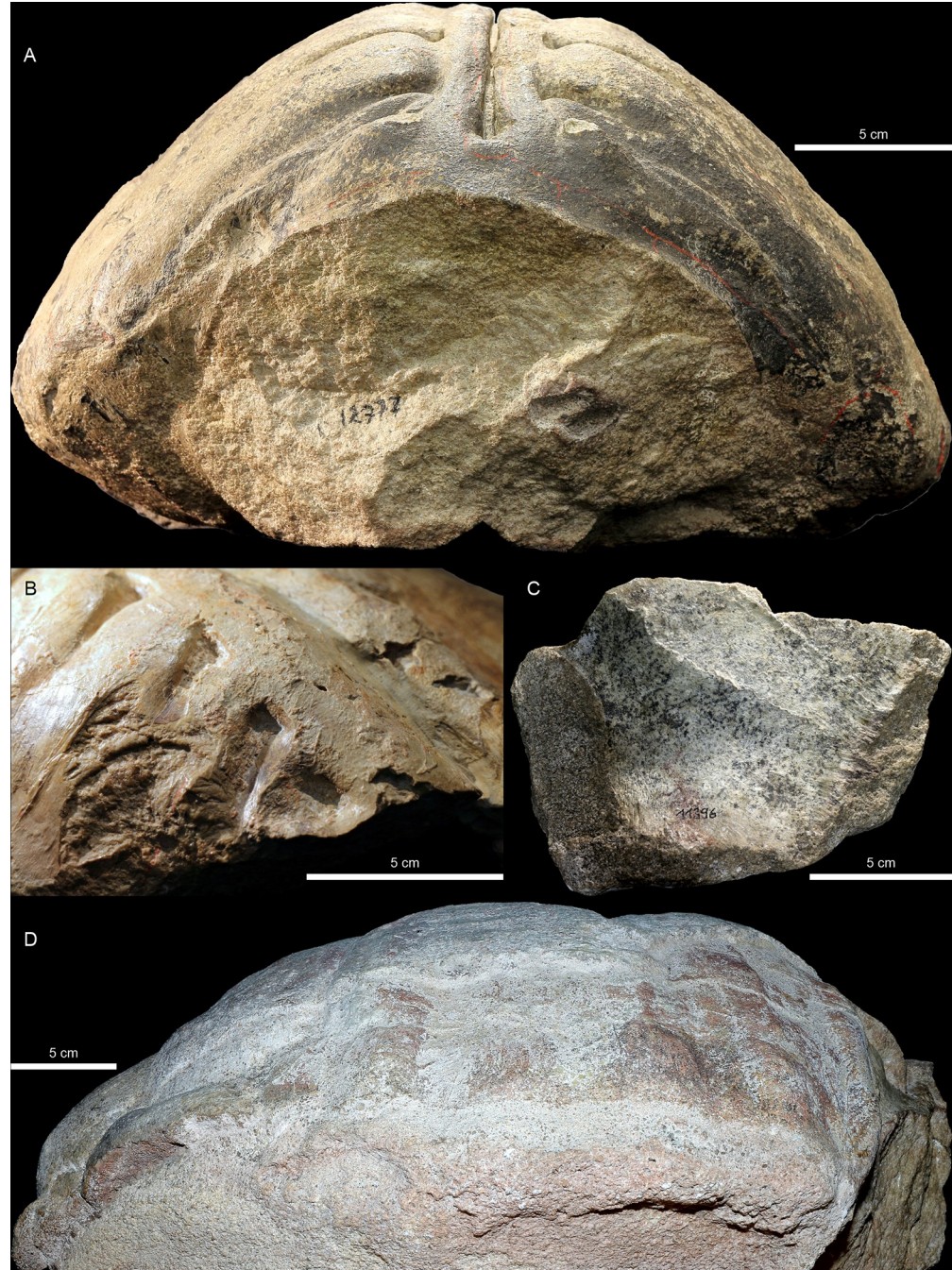

**Fig 3. *Proterochersis robusta* steinkerns from the Lower Löwenstein Formation. A**, SMNS 12777, anterior view. Note the distinct grain of the light grey sandstone matrix and the less coarse, darkened lining of the surfaces formerly adjacent to the internal surfaces of the shell bones. Bright red lines are artificial, and their layout is not representative of bone sutures. **B**, **C**, SMNS 11396, fragmented steinkern: (**B**) posterior left region of the carapace in oblique, posterolaterodorsal view, showing smooth surfaces possibly representing remains of the visceral bone cortex; (**C**) plastron region in ventral view, showing the impression of the visceral surface of the plastron with no bone residue but with a lining of very fine sediment, capturing minor detail, such as suture lines. **D**, CSMM uncat., lateral right view, showing remaining shell bone fragments (blue-grayish in the dorsal and posterior part, note the reddish cross section), imprint of the visceral part of the carapace (note the reddish tint), and coarse matrix exposed on the break surface in the ventrolateral part of the specimen.

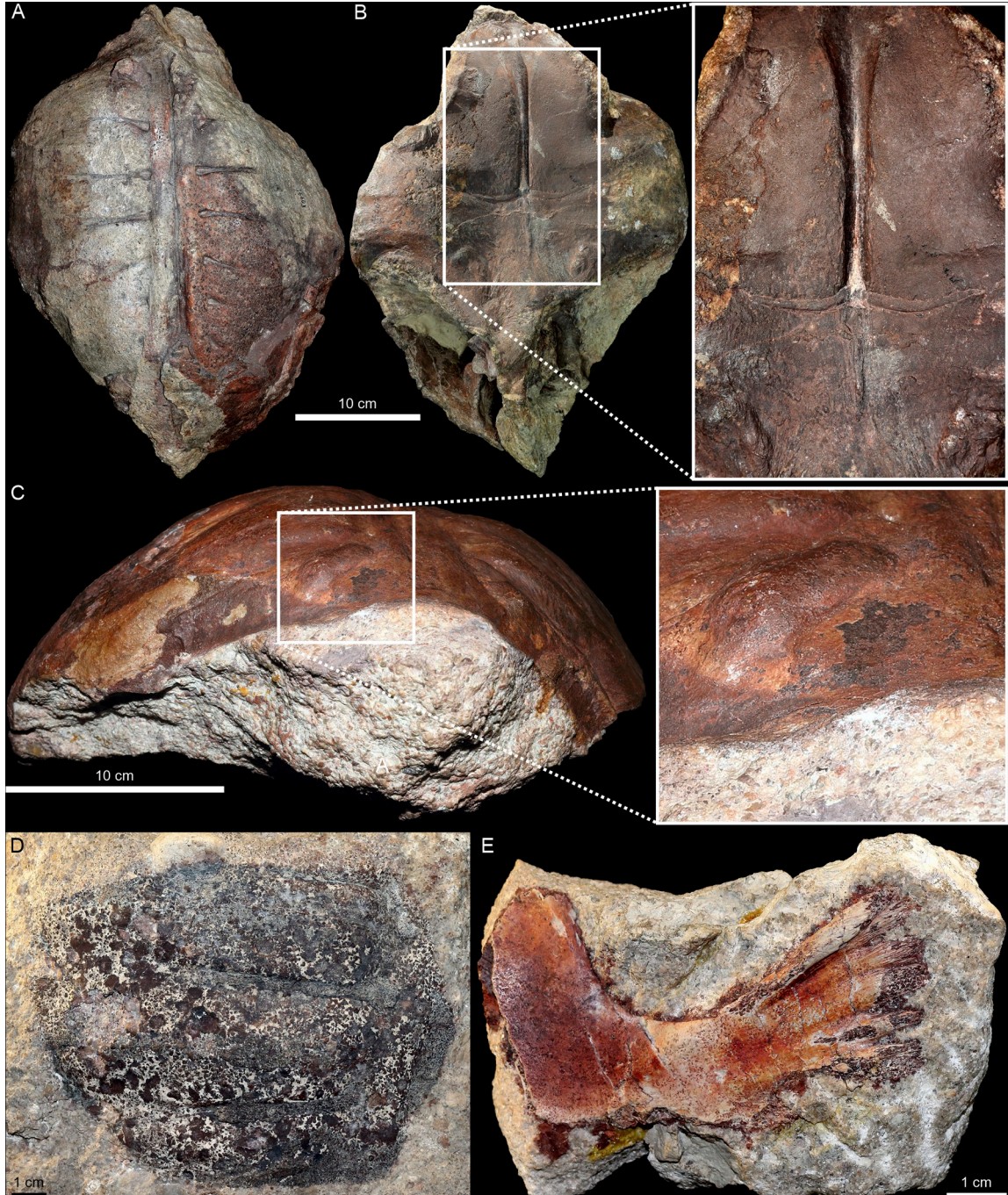

**Fig 4. *Proterochersis robusta* steinkerns and fragments from the Lower Löwenstein Formation. A**, **B**, SMNS 16603, steinkern in (**A**) dorsal and (**B**) ventral view with closeup of the middle of the plastron imprint. Note the reddish-gray color of the remaining bones (mostly the posterior right part of the carapace and right side of the pelvis), reddish matrix areas in the carapace region (including dark red clasts), and dark red surface of the plastron imprint preserving fine anatomical detail. **C**, SMNS 17930, steinkern in anterior view with closeup of the anterior right part of the carapace region. Note the flaking reddish surface of the bone imprint and sharp border with the lighter, coarse matrix. **D**, SMNS 17756, an impression of the visceral surface of the carapace fragment. Note the dark residue of the bones, dark clasts in the area of the imprint, and little to no discoloring of the surrounding cement. **E**, SMNS 81917, plastral bone fragment in visceral view. Note that the bone is beige with reddish tinting, the surrounding matrix is partly discolored, and dark red infills of the intertrabecular areas.

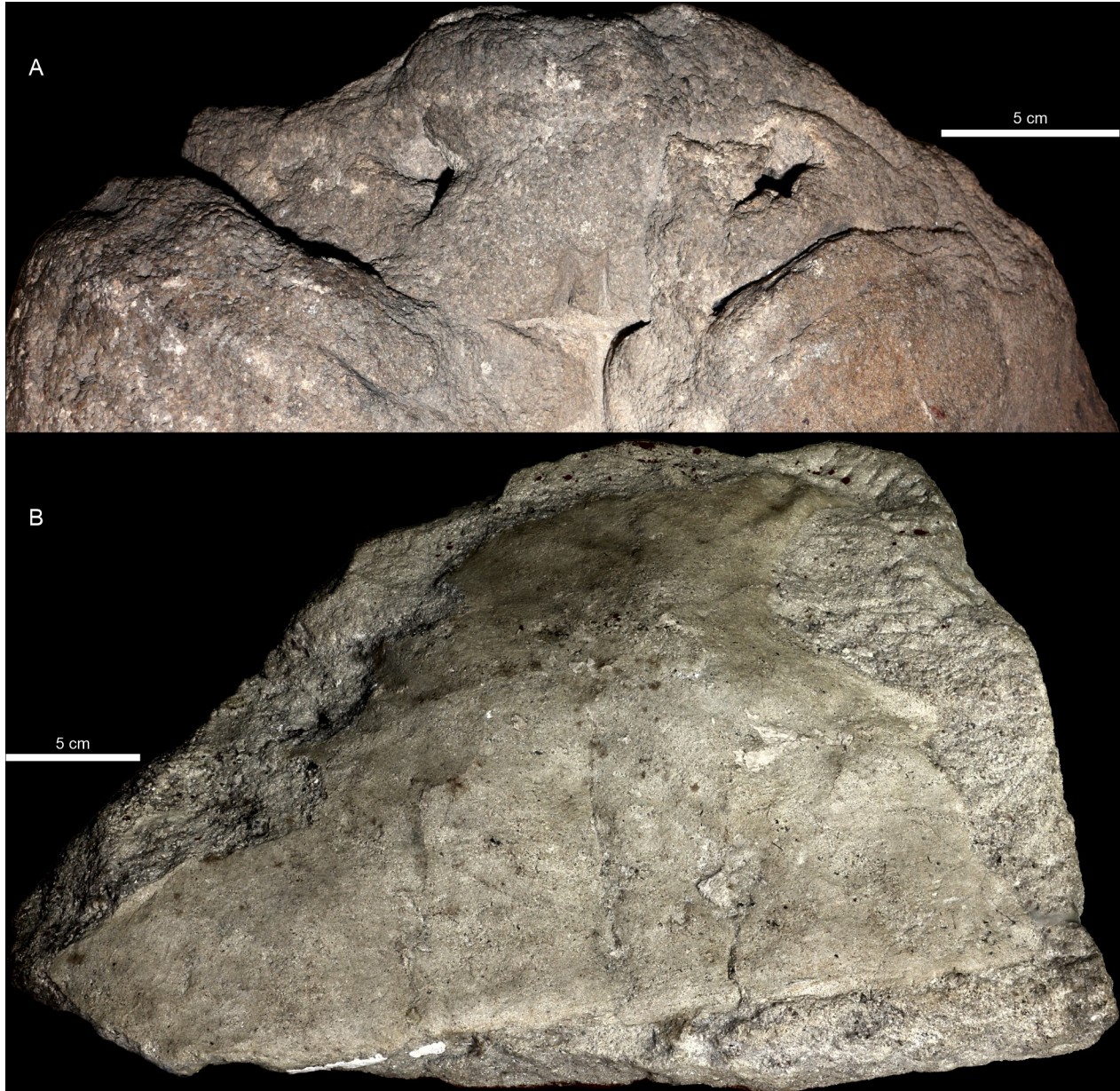

**Fig 5. *Proganochelys quenstedtii* from the Middle or Upper Löwenstein Formation. A**, GPIT-PV-30000, anterior part of the steinkern in dorsal view. Note the rough surface with distinct coarse grain of the sandstone matrix. **B**, SMNS 10012, imprint of the visceral surface of the plastron in dorsomedial (slightly oblique) view. Note the finer structure of the imprint compared to the surrounding matrix and no discoloration.

the Middle or Upper Löwenstein Formation (GPIT-PV-30000, Fig 5A), as well as some partial steinkerns of *Proterochersis robusta* from the Lower Löwenstein Formation (SMNS 16668, SMNS 17756; Fig 4D) have very rough surfaces, with a well-visible, coarse grained whitish sandstone and poor retention of fine details. This, however, in addition to the coarseness of the sediment, may be an effect of weathering due to their prolonged exposure to environmental factors [17, 18]. However, in most cases the surfaces of the imprints from the Löwenstein Formation preserve very fine detail. That is especially clear in the case of specimens such as *Proterochersis robusta* SMNS 17930 (Fig 4C), in which most of the surfaces adjacent to the

removed carapace bones have a very fine and uniform grain size and are reddish-brown in color, whereas the sediment at the core of the steinkern is a coarse-grained whitish sandstone. The surface of *Proterochersis robusta* SMNS 11396 partial steinkern from Rohracker locally preserves an excellent detail (Fig 3C) and smoothness (Fig 3B), but it seems that this effect was achieved by retention of a thin layer of the visceral bone cortex on the surface of the stone core during preparation [13].

In some cases, particularly in the specimens from Murrhardt, the rock matrix just adjacent to the bone and filling the intertrabecular spaces becomes discolored (usually reddish), hinting at distinct conditions forming at the interface between the fossilizing bone and its environment. This phenomenon is well observed in *Proterochersis robusta* SMNS 81917 (Fig 4E), a juvenile plastral bone from that locality still embedded in the matrix: the surrounding rock as well as the bone are light beige to gray, but the bone surface bears an orange-red, uneven stain, and the grains or clasts just adjacent to it, as well as the contents of the intertrabecular spaces are dark red. The cement between the grains or clasts of the matrix seems to be less affected. In *Proterochersis robusta* SMNS 17930 from Oberbrüden, the bone-adjacent surfaces of the steinkern are also discolored (Fig 4C), but in that specimen the change of color to reddish-brown applies to the bone as well, while the intertrabecular spaces are infilled by lighter, more grayish sediment.

In the case of SMNS 15479, most of the slab consists of coarse sandstone but the plastron imprint is lined with a less than 1 mm thick layer of finer-grained sediment (Fig 2D). The surfaces of the imprint are mostly light grey but locally darkened (mostly in the anterior and central part) and the matrix around the imprint, exposed by partial erosion or damage to the imprint surface, as well as filling the cracks and scute sulci, is mostly distinctly darker, reddish to brown. The darkest patch is in the posterior right part of the specimen, where the imprint is damaged, and the discoloration affects the layers underlying and overlying the surface of the imprint (although note that there are no distinct bedding planes in the sandstone and no information about the exact orientation of the specimen in the matrix is recorded). The left side of the slab is broken, providing a natural cross-section through the imprint surface and the underlying sediment. In that region, patches of ochre to brown discoloration extend into the matrix for about 4 to 35 mm away from the imprint surface, but at the level of the abdominal scutes, the discoloration is nearly absent.

Overall, the rock matrix of the Löwenstein Formation seems to promote the formation of natural shell molds thanks to its relative homogeneity and small grain size, cohesiveness, and relative easiness of separation from the bone. A potential impact of the used recovery and preparation techniques, obviously, cannot be refuted.

## Grabowa Formation

Although the potential of most other turtle-bearing formations for the preservation of natural molds of shells may be underestimated due to collector bias and/or small sample size, these factors seem to not be a viable explanation of the lack of natural molds in the Grabowa Formation of Poland–another formation which produced hundreds of specimens attributable to the genus *Proterochersis* (*Prot. porebensis* and *Prot.* cf. *porebensis*) [14, 38, 62–65]. Despite of the presence of closely related turtle taxa and a somewhat similar lithology as in the Löwenstein Formation, suggesting some similarity of the depositional environment, as well as despite the wealth of fossil, no natural shell molds were ever found in the Grabowa Formation. Two localities of the formation were thus far described which produced numerous turtle fossils: Poręba [53] and Kocury [65].

In the locality of Poręba, turtle fossils are found predominantly in interdigitating layers of gray carbonate mudstone and claystone and light gray to yellowish carbonate conglomerate

(Fig 6A–6D) [14, 38, 53, 62]. The depositional setting was characterized as a fluvial system (possibly braided or anastomosing river system) of variable over time, high (conglomerates) to low (mudstones) energy [14, 38, 64]. Locally, the claystone grades into harder, gray or ochre-colored sandstone. In the mudstone and claystone matrix, the bones are black and usually well-preserved, and can be found either embedded directly in the relatively soft matrix, which may be easily removed with water and brush (especially when the specimens are small), or partly or fully covered with a harder, more lithified gray calcareous crust (Fig 6C). This crust is hard enough to be resistant to removal without the use of tools, but relatively brittle and very easy to separate from the bones in small fragments with the help of the preparation equipment, without damage to the specimen. The conglomerates are significantly harder and more diffi-cult to prepare, and the bones contained inside are usually preserved in a worse condition and lighter brown, sometimes with an orange tint. Due to the hardness of the conglomerate and its strong connection to the bone, it is often difficult to prepare mechanically without damaging the surface of the specimen–irregular pieces of the rock often tend to separate together with a layer of cortical bone. Rarely, the bones are closely associated with oncoids (Fig 6D) [38, 64]. Bones exposed to environment for an extended time tend to lose color, becoming bluish, gray, or even nearly white. There is no noticeable discoloration of the sedimentary matrix next to bone.

In Kocury, turtle remains were also found in gray or yellowish conglomerates (Fig 6E) interdigitated with layers or apparently bone-barren claystone and mudstone [65]. A prelimi-nary, two-step scenario of the conglomerate formation in that locality, consisting of an infilling of channels of fluvial origin with gravel, and its subsequent lithification by calcitic cement in a caliche-like fashion, was proposed by Czepiński et al. [65]. Bones found in the locality are dark brown to dark beige, sometimes with a reddish tinting, which may also affect the adjacent matrix. The preservation state of their surfaces is usually better than in the conglomerate-embedded bones from Poręba but not as good as in some mudstone- or claystone-embedded bones from the latter locality. They are, however, mostly incomplete, and usually represent only small fragments. Only one specimen from the locality, a partial osteoderm of the aetosaur *Kocurypelta silvestris* Czepiński et al., 2021 [65] ZPAL V. 66/19, is preserved embedded in the matrix with a part missing (most probably destroyed during recovery), revealing a natural imprint of the external surface (Fig 6E) [65]. While this presents some potential of the Gra-bowa Formation conglomerates to form natural molds of skeletal specimens and the general outline of the bone and basic features (anterior bar) are distinguishable, the retention of sur-face detail in that case is generally poor. Moreover, the potential to produce internal or external molds of turtle shells seems to be hampered by the depositional setting promoting preservation of mostly small, broken bone fragments over more complete specimens.

## Results

### Morphological description

The natural mold, SMNS 15479, captures most of the right side of the anterior plastral lobe (the gular and extragular processes appear incomplete) and complete left side of the posterior plastral lobe, with partial preservation of the opposing sides. The imprint of the central part of the plastron is also preserved, but the bridge on both sides is poorly and incompletely impressed. Overall, impressions of all the scute outlines are at least partially recognizable and areas of most of them are represented in full at least on one side of the body (Fig 7C, Table 2). The preserved parts of the scute area imprints are mostly delineated by relatively clear sulci, but in some cases, these sulci are partially obscured by cracks or either poor preservation of surface of the mold or some additional material (e.g., clasts) which initially separated the

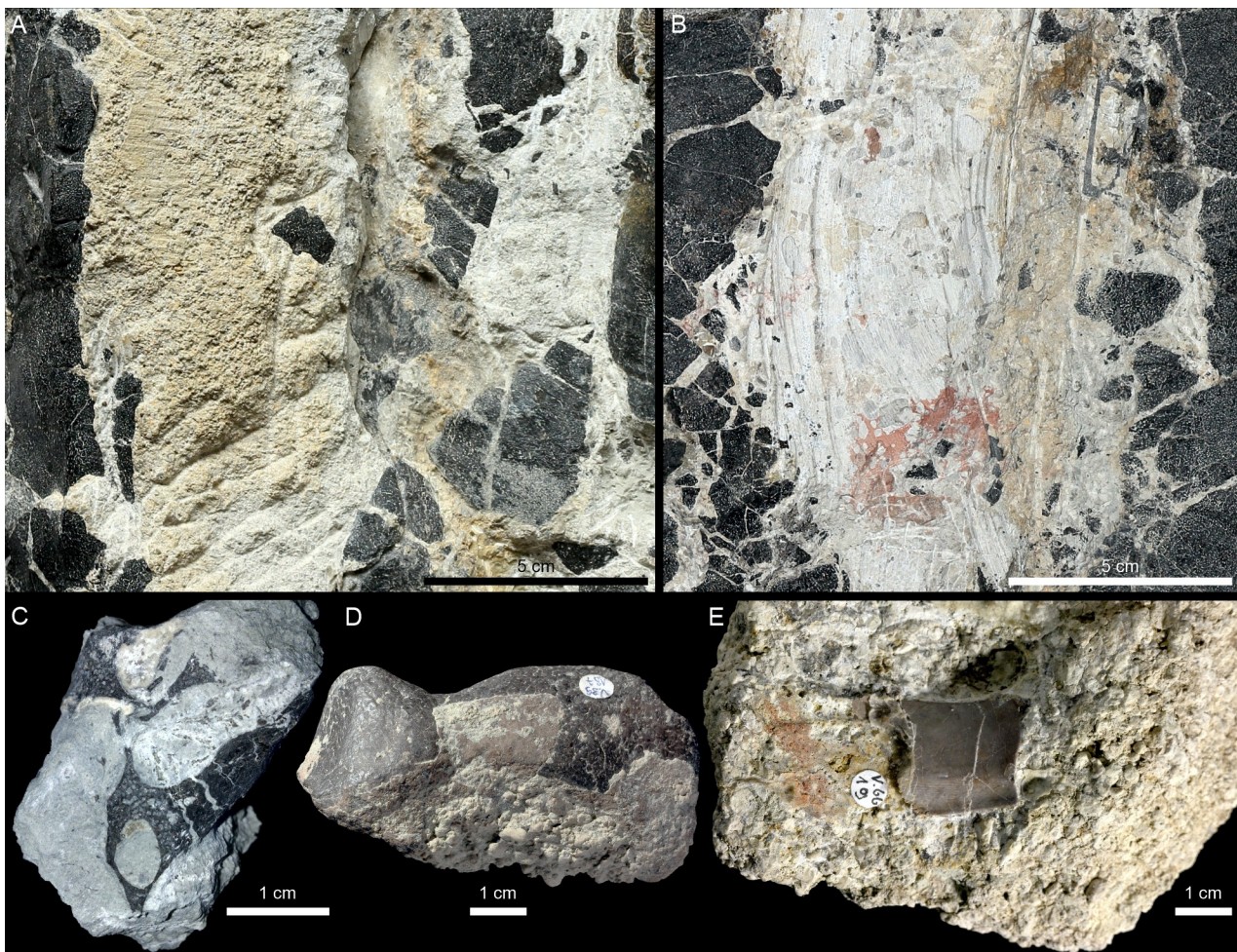

**Fig 6. Bone material and associated rock matrix from the Grabowa Formation. A**, **B**, *Proterochersis porebensis* ZPAL V. 39/34, (**A**) closeup of the dorsal side of the specimen, showing black, fractured bone in the grey-yellowish sandy-claystone matrix and (**B**), closeup of the ventral side of the specimen presenting the conglomerate matrix with variable grey, yellowish, to red coloration. Note that usually bones found in the conglomerates of the Grabowa Formation are brown rather than black. **C**, *Proterochersis porebensis* ZPAL V. 39/49, vertebral column and rib fragments (black) embedded in a lithified grey mudstone crust. **D**, *Proterochersis porebensis* ZPAL V. 39/187, gular part of the plastron (dark brown) with residue of grey mudstone with larger clasts and an oncoid-like layer. **E**, *Kocurypelta silvestris* ZPAL V. 66/19, part of a light brown osteoderm in visceral view embedded in a conglomerate matrix. Note that a part of the bone is missing, revealing an external mold with poor retention of surface detail (on the left).

surface of the plastron from the imprinted surface and were subsequently lost. However, in some areas, particularly in the central region of the plastron, the surface preservation is surprisingly good, and some fine details can be seen (Fig 7A, 7B, 7D–7F). These, most notably, include scute growth marks of the pectoral scutes and fine, oblique striation on the right pectoral scute and abdominal scute areas.

Because the virtual positive (Figs 7 and 8A) is significantly easier in interpretation than the original fossil (negative) and represents the initial geometry of the plastron, the description is presented as if referring to the plastron itself, although the features on the imprint are inverted. Overall, the plastron of SMNS 15479 presents a normal proterochersid morphology (Fig 8A–8D). While the three-dimensional conformation of the surface might have been impacted by differential compactness of the sediment during diagenesis, based on comparisons of other specimens of Triassic turtle plastra and usual lack of significant deformation in the turtle fossils

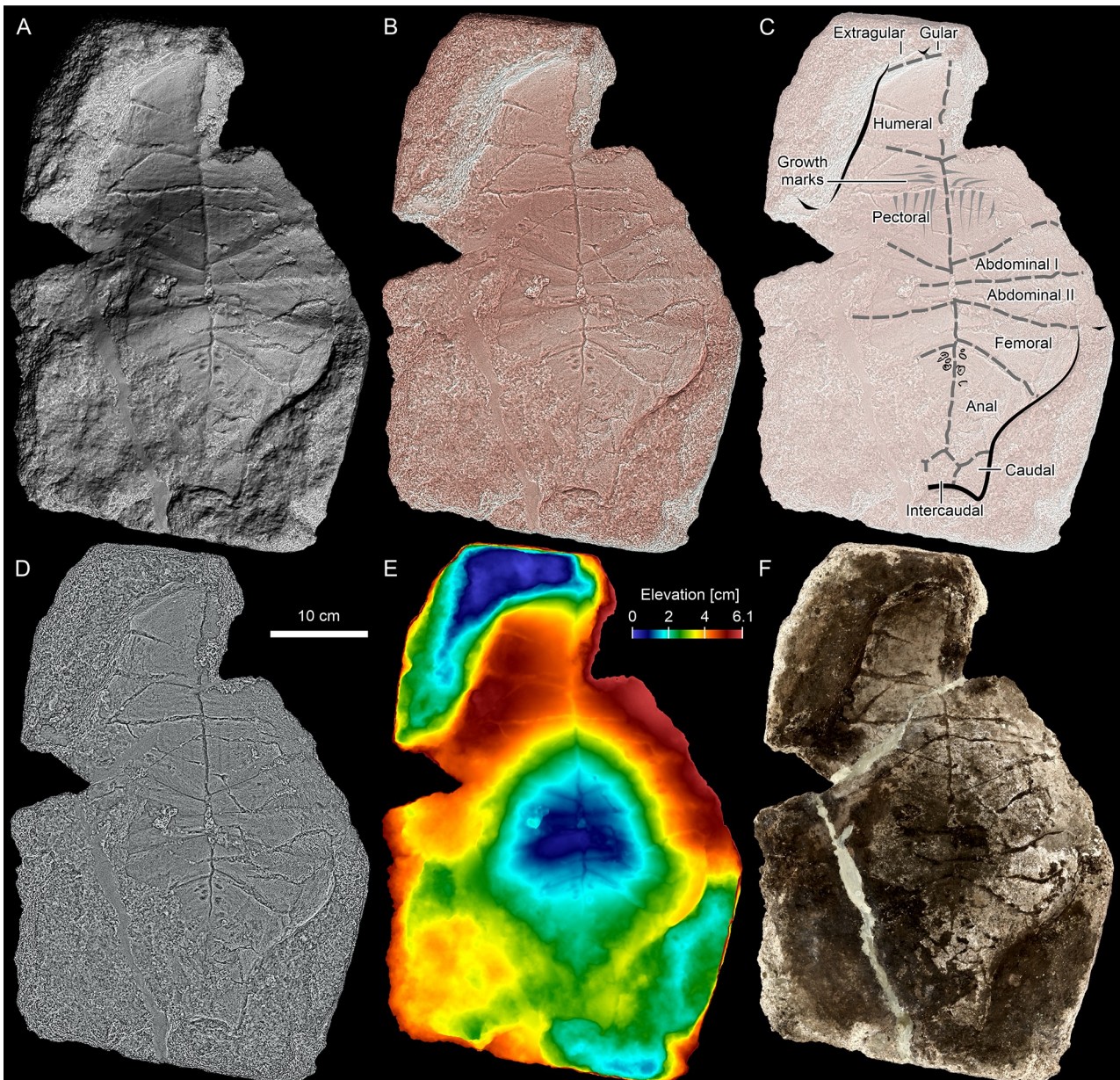

**Fig 7.** ***Proterochersis robusta*** **SMNS 15479, virtual positive. A**, Lambertian and **B**, Lit Sphere Radiance Scaling, presenting the geometry of the plastron. **C**, interpretative drawing showing the scute layout. **D**, Grey Descriptor Radiance Scaling ignoring lighting and geometry and highlighting edges. **E**, depth map presenting elevation; note the central concavity. **F**, vertex color with shading disabled, presenting the coloration of the specimen. All panels in orthographic view.

from the Löwenstein Formation, we consider it at least mostly original. There is a distinct concavity in the central part and most of the anterior lobe and the bridge region is ventrally convex (Fig 7E), which is typical for *Keuperotesta limendorsa* (Fig 8D) and *Proterochersis* spp. (Fig 8B and 8C) The plastral curvature is much weaker in *Odontochelys semitestacea* (Fig 8F), the Australochelyidae (*Palaeochersis talampayensis*, Fig 8E, and *Waluchelys cavitesta* Sterli et al., 2021 [33]), and *Proganochelys quenstedtii* (Fig 8G and 8H) [13–16, 18, 25, 28, 32–34, 36, 37, 50]. The gulars and extragulars, although not preserved in full, were clearly located dorsally

**Table 2. Scute area preservation in SMNS 15479.**

| Scute | Left | Right |
|---|---|---|
| Gular | Not preserved | Partially preserved |
| Extragular | Not preserved | Partially preserved |
| Humeral | Partially preserved | Preserved |
| Pectoral | Partially preserved | Preserved |
| Abdominal I | Partially preserved | Partially preserved |
| Abdominal II | Partially preserved | Partially preserved |
| Femoral | Preserved | Partially preserved |
| Anal | Preserved | Partially preserved |
| Caudal | Preserved | Partially preserved |
| Intercaudal | Preserved | |

relative to the external surface of the humeral scutes. This, as well, is a feature common to adult proterochersids (Fig 8B–8D) but not juveniles [14–16, 20, 38] whereas in *Odontochelys semitestacea* (Fig 8F), *Proganochelys quenstedtii* (Fig 8G), the Greenland proganochelyid NHMD 190349, and the australochelyids (Fig 8E), the anterior plastral scutes are almost level with scutes or only slightly dorsal to the anterior surfaces of the humeral scutes [14–16, 18, 25, 32, 33]. The presence of both the gular and extragular scute pair, although very poorly indicated, further differentiates SMNS 15479 from the australochelyids (Fig 8E) [31–34]. The poor preservation of the extragulars makes it difficult to evaluate the lateral extent of the extragular projections, but as preserved there is no evidence of their lateral extension beyond the antero-lateral edge of the humeral scutes. If true, this would distinguish SMNS 15479 from *Odontochelys semitestacea* (Fig 8F), *Proganochelys quenstedtii* (Fig 8G), '*Proganochelys*' ruchae, and the Greenland proganochelyid NHMD 190349 [18, 20, 25, 28, 37, 39, 41], although this character is variable in the Proterochersidae (Fig 8B–8D) [14, 15, 18, 22]. The anterior plastral projections seem to not extend laterally in the australochelyids (Fig 8E) [31–34].

The humeral scutes were large and trapezoid. They were nearly as wide as they are long, as is typical for the proterochersids and *Odontochelys semitestacea* (Fig 8F), but in contrast to *Proganochelys quenstedtii* (Fig 8G), in which they are narrower and more elongated [14–16, 18, 22, 25, 28, 37]. The morphology in the australochelyids is uncertain due to the poor distinctiveness of the scute sulci in those turtles (Fig 8E), but is seems to be more similar to the state in *Proganochelys quenstedtii* rather than the proterochersids [31–34]. The distinct, oblique (anterolaterally facing) step in the anterior part of the humeral scute is more exaggerated than in most other proterochersid specimens, although a similar feature is present in *Proterochersis porebensis* ZPAL V. 39/385 and the holotype of '*Proganochelys*' ruchae (SM2015-1-001, former TF 1440–6), and less marked anterolateroventrally directed faces are also present, e.g., in *Proterochersis porebensis* ZPAL V. 39/48, ZPAL V. 39/49 (Fig 8C), ZPAL V. 39/387, as well as in *Proterochersis robusta* SMNS 17561 (Fig 8B) [15, 20]. In *Proterochersis porebensis* ZPAL V. 39/379 [15] and *Keuperotesta limendorsa* SMNS 17757 (Fig 8D), this region takes the form of a subtriangular, in ventral view posteromedially gently concave and delineated by a subtle groove (ZPAL V. 39/379) or ridge (SMNS 17757), flat surface, instead. The humeropectoral sulci are directed significantly more laterally than anteriorly, which is also observed in the proterochersids (Fig 8B–8D) and *Waluchelys cavitesta* but different than the more acute, clearly V-shaped layout in *Odontochelys semitestacea* (Fig 8F) and *Proganochelys quenstedtii* (Fig 8G) [14–16, 18, 22, 25, 28, 37]. The lateralmost ends of the humeropectoral sulci are obscured by a crack, so the presence of their anterior deflection cannot be confirmed. In contrast to *Proganochelys quenstedtii* (Fig 8G) and *Keuperotesta limendorsa* (Fig 8D) but in line with *Odontochelys*

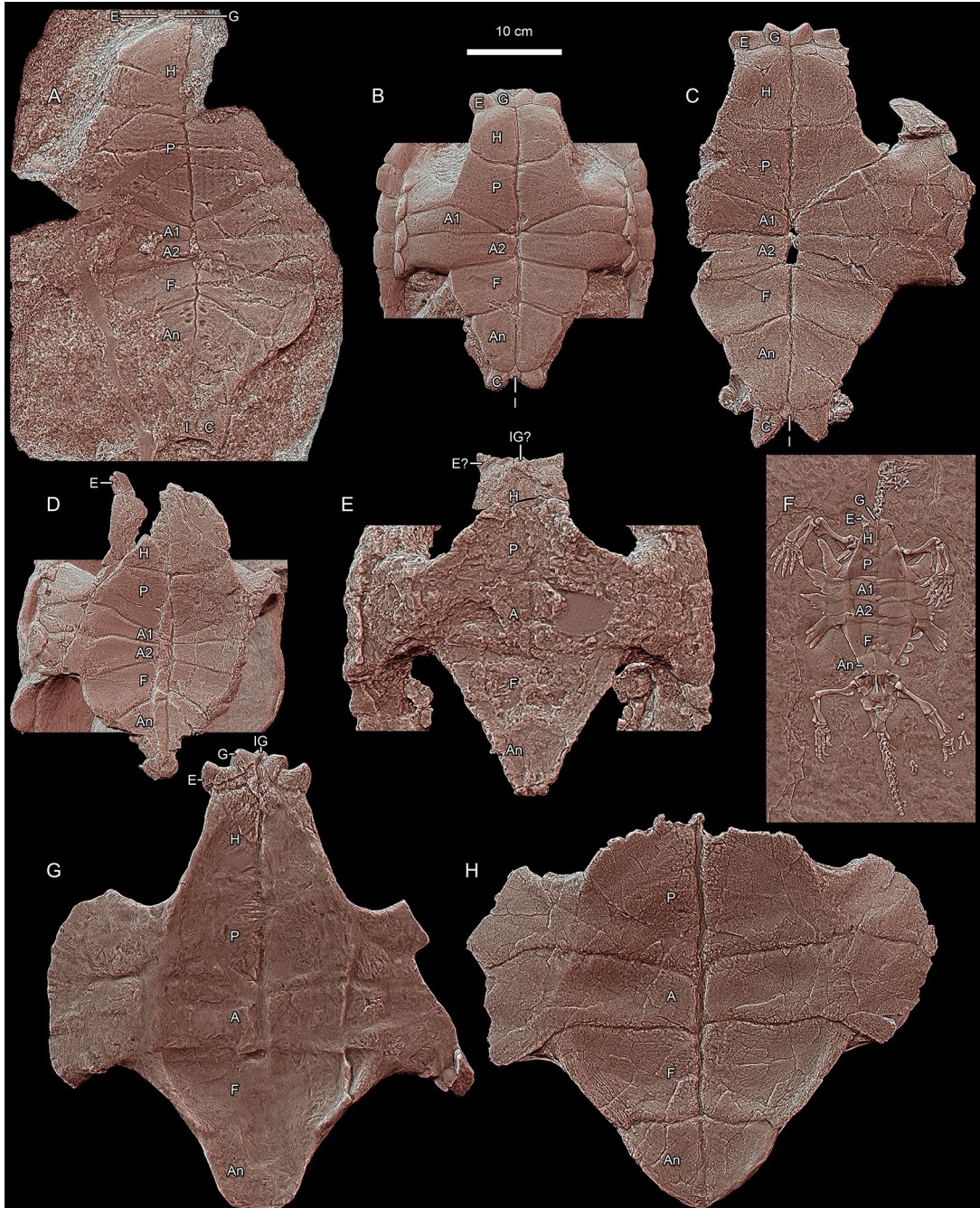

**Fig 8. Morphology of Triassic stem turtle plastra. A**, *Proterochersis robusta* SMNS 15479 (Germany). **B**, *Proterochersis robusta* SMNS 17651 (Germany); note that the lack of medial contact of the first pair of abdominal scutes is not typical for the species. **C**, *Proterochersis porebensis* ZPAL V. 39/49 (Poland). **D**, *Keuperotesta limendorsa* SMNS 17757 (Germany). **E**, *Palaeochersis talampayensis* PULR 068 (Argentina). **F**, *Odontochelys semitestacea* IVPP V 13240 (China). **G**, *Proganochelys quenstedtii* SMNS 16980 (Germany). **H**, *Proganochelys quenstedtii* MB.1910.45.2 (Germany); note that the tapered posterior edge most likely is an effect of restoration rather than an actual morphological feature. Carapaces partially digitally removed in B, D, and E for clarity. Abbreviations: A, abdominal scute; An, anal scute; C, caudal scute; E, extragular scute; F, femoral scute; G, gular scute; H, humeral scute; I, intercaudal scute; IG, intergular scute; P, pectoral scute. All panels in orthographic view with the Radiance Scaling (Lit Sphere) shader enabled.

*semitestacea* (Fig 8F) and *Proterochersis* spp. (Fig 8B, 8C), the posteromedial limit of the humeral scute is located significantly anteriorly to the axillary notch [14–16, 18, 22, 25, 37].

The pectoral scute was about as long as the humeral scute and its main surface, located on the anterior lobe of the plastron, is almost as long as it is wide. Both pectoral scute areas bear shallow but clear L-shaped (the bend directed anteromedially) scute growth marks on the pectoral scute areas (Figs 7A–7D, 8A). Although relatively common on the carapaces, bent scute growth marks are generally not detectable on the plastra of proterochersids (Fig 8B–8D) [14–16] and are also absent in other Triassic turtles (Fig 8E, 8G, 8H). The pectoroabdominal sulci are slightly asymmetrical, the left one being shifted about 8 mm anteriorly relative to the right one, and set at a more acute angle relative to the midline than the humeropectoral sulci. Note that although Gaffney [18, 37] in his reconstructions presented the pectoroabdominal scute of *Proganochelys quenstedtii* as nearly straight and transverse, which was later copied by numerous other authors [16, 22, 34], this morphology is in fact not represented by any of the actual specimens (Fig 8G, 8H) [18, 36, 37]. The surface of the posterior part of the right pectoral scute area as well as the surface of the first right abdominal scute area are gently striated, creating a subtle herringbone-like pattern along the right pectoroabdominal sulcus. A fine striation associated with plastral scutes is visible in *Proterochersis robusta* SMNS 17755 and (much less distinct) in SMNS 18440, albeit in both cases it is more perpendicular to the pectoroabdominal sulci [16].

The sulci in the central part of the plastron delineate two pairs of short, wedge-shaped and medially meeting abdominal scutes. The first right abdominal is about 8 mm shorter than the first left abdominal, the sulci between the first and second abdominal pair are virtually level and directed almost completely laterally, although on the left, the sulcus is inclined slightly more anteriorly. This fits within the normal variability of the abdominal scutes observed in proterochersids (Fig 8B–8D) [14–16]. SMNS 15479 shares the presence of two pairs of the abdominal scutes with *Odontochelys semitestacea* (Fig 8F) and the Proterochersidae (Fig 8B–8D), but not with *Proganochelys quenstedtii*, which had only a single pair of the abdominal scutes (Fig 8G, 8H) [13–16, 22, 25, 34]. The abdominofemoral sulci are directed predominantly laterally and gently posteriorly, normally for the proterochersids but distinguishing SMNS 15479 from *Odontochelys semitestacea* and *Proganochelys quenstedtii*, both having virtually completely laterally directed abdominofemoral sulci (Fig 8F–8H) [13–16, 18, 22, 25, 28, 36, 37].

The femoral scutes were wider than long, gently increasing in length laterally but decreasing in width posteriorly, with distinctly convex free posterolateral edges. The femoroanal sulci are directed posterolaterally at a slightly more acute angle than the abdominofemoral sulci. The bone in the region of the femoral scutes was clearly thick, as indicated by the depth of the imprint. All of the above characters are typical for the Proterochersidae (Fig 8B–8D) [13–16, 22]. *Odontochelys semitestacea*, adult *Proganochelys quenstedtii*, and the australochelyids lack the distinct, bulge-like convexity (present, however, in the juvenile *Proganochelys quenstedtii* SMNS 17203) and thickening of the femoral scute edge [18, 26, 28, 31–37]. The inverted V-shaped femoroanal sulci are furthermore shared with *Palaeochersis talampayensis* (Fig 8E) [31, 32]. The morphology in *Odontochelys semitestacea* is poorly defined–although Lyson et al. [28] presented the sulci as mostly transverse, a more posterolateral inclination seems more plausible (pers. obs., IVPP V 13240; Fig 8F). In contrast, Gaffney [18, 37] illustrated the femoroanal sulci of *Proganochelys quenstedtii* as posterolaterally inclined, but the sulci are not traceable in SMNS 16980 (Fig 8G), and very poorly traceable in SMNS 17204. The latter specimen, together with MB.1910.45.2, in which the sulci are very clear (Fig 8H), suggests that they were directed laterally throughout most of their length, only with a slight anterior deflection medially and slight posterior deflection laterally (photographs of MB.1910.45.2 plastron in Jaekel [36] and

Gaffney [18], pers. obs.; note that the drawing of MB.1910.45.2 plastron in Jaekel's [36] Fig 5 is not fully accurate). As in the case of the pectoroabdominal sulci, the left femoroanal sulcus is about 8 mm anterior relative to the right.

The anteromedial edges of the anal scutes extended far anterior to the posterolateral edges of the femoral scutes, so the anal scute area has its widest point at the level of the lateral end of the femoroanal sulcus. In contrast to *Odontochelys semitestacea* (Fig 8F) and *Proganochelys quenstedtii* (Fig 8G, 8H), the anal scutes of SMNS 15479 had gently concave rather than straight or gently convex posterolateral edges and were proportionally longer [18, 25, 28, 37]. Their proportions are more similar to the australochelyids (Fig 8E), but in the latter the posterolateral edges are straight [31–34]. The profile of those scutes in *Proterochersis* spp. is variable, from gently convex, through sinuous, to gently concave posterolaterally (Fig 8B, 8C) [13–16, 22]. Their edges are not preserved in *Keuperotesta limendorsa* (Fig 8D) [14]. The anal scute areas of SMNS 15479 bears anteromedially a cluster of oval pits, expressed on the surface of the original slab as small (diameter about 4–6 mm) convex, relatively smooth protrusions of similar coloration and grain to the surrounding matrix (Fig 9). The origin of those pits is uncertain, but they may represent shell pathology.

SMNS 15479 has the paired caudal and single intercaudal scute, thus far only known in *Proterochersis* spp. (Fig 8B, 8C) [13–16, 18, 22]. In *Keuperotesta limendorsa*, the posterior edge of the plastron is not preserved (Fig 8D) [14]. In *Odontochelys semitestacea* (Fig 8F), *Proganochelys quenstedtii* (Fig 8G), and the australochelyids (Fig 8E), the plastron ends posteriorly in as straight or gently rounded edge [18, 25, 28, 31, 33, 37, 81]. The size and morphology of those scutes in *Proterochersis* spp. is highly variable [15], as is the depth of the surrounding sulci, and SMNS 15479 fits within this observed variability.

There is little information preserved about the bridge, but the inguinal notch seems to be located far posterior to the posteriormost point of the humeropectoral sculcus, distinguishing SMNS 15479 from *Keuperotesta limendorsa* SMNS 17757 (Fig 8D) and *Proganochelys quenstedtii* (Fig 8G) [14, 18, 37]. There is no imprint of the lateral parts of the bridge, so the presence of inframarginal and axillary scutes cannot be observed.

## Size

As preserved, the maximum length of the plastron imprint of SMNS 15479 is 45.6 cm, the midline length is 44.4 cm (43.9 cm excluding the incomplete gulars, i.e., measuring from the gular-humeral scute sulcus to the posterior edge of the intercaudal scute), and the maximum width (including the poorly preserved and incomplete imprints of the bridge) is about 33.0 cm. The width of the posterior plastral lobe, at the level of the widest part of the femoral scute, just posterior to the inguinal notch, was about 25.4 cm (based on doubled distance between the lateralmost point of the single fully preserved left femoral scute area and the midline, indicated by the mesial sulcus, i.e., 12.7 cm). This is extremely large for a Löwenstein Formation proterochersid and in line with the largest individuals of *Proterochersis porebensis* from Poland. The holotype of *Keuperotesta limendorsa* SMNS 17757 and *Proterochersis robusta* SMNS 16442 are the largest skeletal proterochersid specimens from Germany (Table 3), although their plastra are incomplete. In SMNS 17757 the preserved part of the plastron extends from the anterior edge of the extragular scute to the level of the intercaudal scute and is 33.5 cm long, with the midline length (as preserved) of 32.5 cm. A corresponding part of the SMNS 15479 imprint is over 40 cm long, making it about 20% larger than SMNS 17757. In SMNS 16442 the plastron is preserved in two parts (anterior lobe and posterior lobe) and supplemented by a natural mold of the inside of the shell (steinkern), but these parts lack overlap complicating the size estimation. Based on the measurements of the corresponding parts of the plastron, SMNS 15479 was

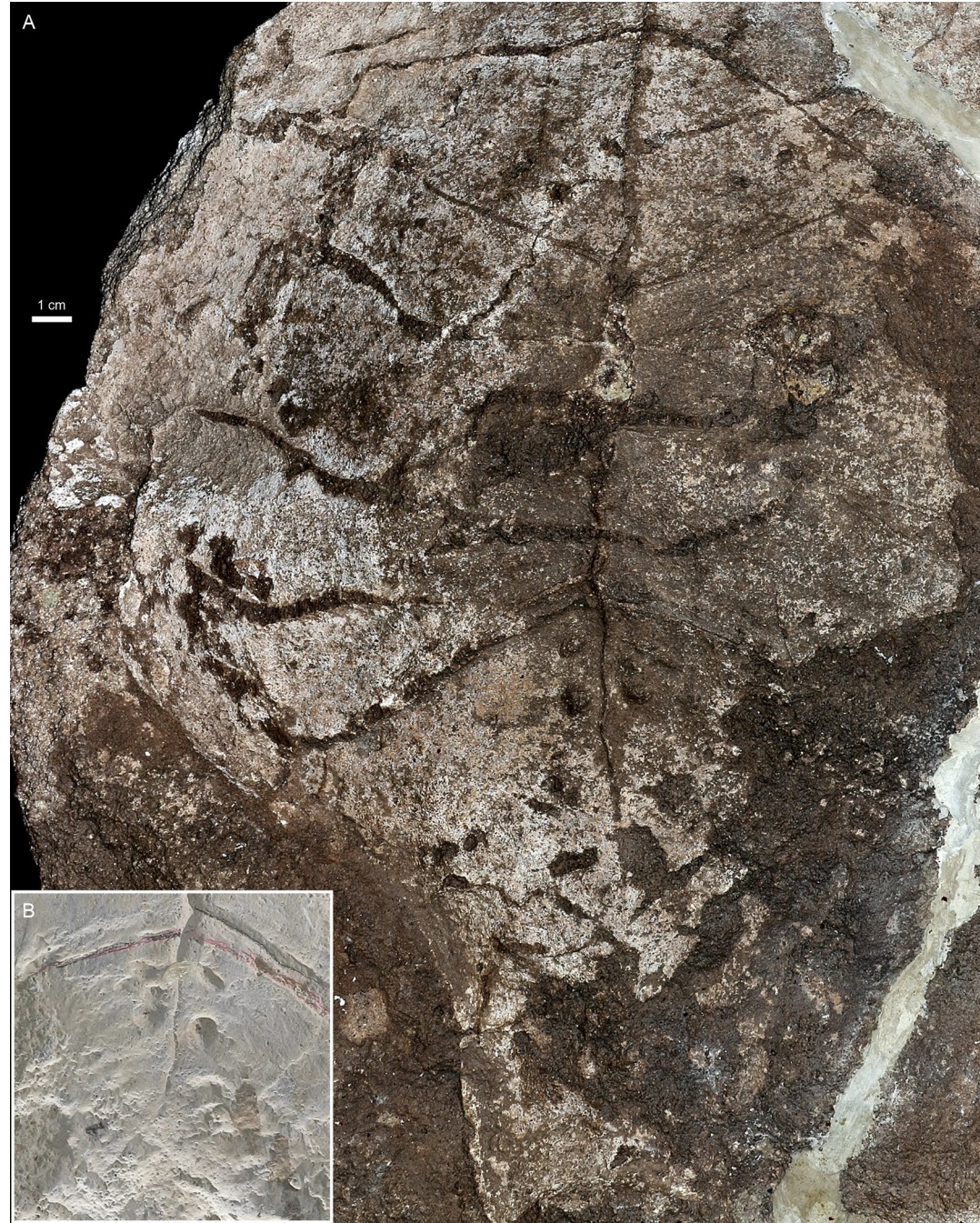

**Fig 9.** *Proterochersis robusta* **SMNS 15479. A**, closeup of the surface of the slab capturing the middle and posterior part of the plastron, and **B**, closeup of the pitted area of the plaster cast.

also roughly 1.2 times the size of SMNS 16442. The plastron length of the holotype of *Proterochersis robusta*, SMNS 12777, is estimated to be about 74% of SMNS 15479, and the estimate for SMNS 17755 is approximately 75.5% of SMNS 15479. The rest of *Proterochersis robusta* specimens fit within the range of 61% (SMNS 16603) to 73% (SMNS 50917) of SMNS 15479 plastral length. *Proterochersis porebensis* is on average larger than *Proterochersis robusta*, with the largest complete specimen (ZPAL V. 39/49) having the midline plastral length of 38.8 cm

**Table 3. Plastron measurements of SMNS 15479 and other proterochersids.** Given plastron lengths exclude the lengths of gulars, extragulars, and caudal projections beyond the posterior edge of the intercaudal scute due to their variable development and incompleteness of the gular imprint in SMNS 15479, i.e., they correspond to the midline length or equivalent if the midline is not fully preserved. Femoral scute widths may be imprecise in SMNS 11396, SMNS 17755, and SMNS 18440 due to the plastra of those specimens being broken about the mesial plastral sulcus and possibly missing some bone medially. Measurements of SMNS 16442 based partially on the two separated preserved plastral lobes and the associated steinkern, which lack sufficient overlap, so they do not represent the whole plastron. SMNS 16603 preserves only small part of the anterior plastral lobe, so the plastron measurements are based on the associated steinkern of an almost complete plastron. The percentages are rounded up to 0.5% to avoid false precision and account for measurement error, specimen surface damage and weathering, etc.

| Species | Specimen | Plastron length as preserved [cm] | Length of a corresponding part in SMNS 15479 [cm] | % of SMNS 15479 size estimated based on the plastron length | Maximum width of the femoral scute [cm] | % of SMNS 15479 size estimated based on the femoral scute width | Ratio between the size percentage relative to SMNS 15479 estimated based on the femoral scute width relative to plastron length |
|---|---|---|---|---|---|---|---|
| *Keuperotesta limendorsa* | SMNS 17757 | 32.5 | > 40 | < 81 | 9.5 | 75 | 0.93 |
| *Proterochersis porebensis* | ZPAL V. 39/34 | ~26.5 | 43.9 | ~60.5 | 7.7 | 60.5 | 1.00 |
| | ZPAL V. 39/48 | 34.5 | 43.9 | 78.5 | 9.5 | 75 | 0.96 |
| | ZPAL V. 39/49 | 38.8 | 43.9 | 88.5 | 10.5 | 82.5 | 0.93 |
| *Proterochersis robusta* | CSMM uncat. | 29.8 | 43.9 | 68 | 8.3 | 65.5 | 0.96 |
| | SMNS 11396 | 9.3 | ~14.8 | ~63 | 7.6* | 60 | 0.95 |
| | SMNS 12777 | 15.6 | 21 | 74 | 8.6 | 67.5 | 0.91 |
| | SMNS 16442 | 8.9 (humeral scute) + ~10 (bridge, steinkern) + 12.2 (posterior lobe) | 10.1 (humeral scute) + ~12.9 (bridge steinkern) + 14.7 (posterior lobe) | 82.5 | 8.9 | 70 | 0.84 |
| | SMNS 16603 | ~26.7 (steinkern) | 43.9 | 61 | 6.6 (steinkern) | 52 | 0.85 |
| | SMNS 17561 | 28.7 | 43.9 | 65.5 | 7.3 | 57.5 | 0.88 |
| | SMNS 17755 | 23.3 | ~30.8 | ~75.5 | 7.5* | 59 | 0.78 |
| | SMNS 18440 | 9.8 | ~14.8 | ~66 | 7.7* | 60.5 | 0.92 |
| | SMNS 50917 | 17 | ~23.3 | ~73 | 8 | 63 | 0.86 |
| | SMNS 56606 | 7 | 11 | 63.5 | – | – | – |
| *?Proterochersis robusta* | SMNS 15479 | 44.4 | | 100 | 12.7 | 100 | 1 |

(88.5% of SMNS 15479), but fragmentary specimens indicate that the species could attain sizes even about 20% larger [15]. Therefore, SMNS 15479 falls within the size range reachable by *Proterochersis* spp. but is exceptionally big even when compared to the averagely larger *Proterochersis porebensis*. Note that development of the gular, extragular, and caudal scutes in *Proterochersis* spp. is variable [15], so the midline lengths of plastra (excluding the gulars) are used here as more comparable between specimens than total lengths.

The plastron width is here indicated by the maximum width of the femoral scute, just posterior to the abdominofemoral sulcus, as a distinct and usually well-preserved region not particularly susceptible to deformation due to the large thickness and relative flatness. In terms of the width of the plastron, SMNS 15479 is proportionally even larger than other proterochersid

specimens, i.e., about 1.34 times the size of *Keuperotesta limendorsa* SMNS 17757, respectively 1.43 times, 1.48 times, and 1.92 times the size of *Proterochersis robusta* SMNS 16442, SMNS 12777, and SMNS 16603, and 1.21 times the size of *Proterochersis porebensis* ZPAL V. 39/49. In all cases but the juvenile *Proterochersis porebensis* ZPAL V. 39/34, the ratio of the estimated size percentages relative to SMNS 15479 based on the femoral scute to the estimated size percentages based on plastral length is below 1 (from 0.78 for *Proterochersis robusta* SMNS 17755 to 0.96 for *Proterochersis robusta* CSMM uncat. and *Proterochersis porebensis* ZPAL V. 39/48; mean for all measured specimens and all *Proterochersis* spp. = 0.9, for *Proterochersis robusta* = 0.88, for *Proterochersis porebensis* = 0.96) indicating that SMNS 15479 is relatively broader than long when compared to nearly all other individuals. This can be potentially explained as a mixed result of positive allometry during ontogeny, measurement error due to not accounting for the full plastral curvature in fragmentary specimens (possibly overestimating their length due to flattening), or simply intraspecific variability. However, in that regard we observed no clear correspondence between that discrepancy and the absolute size of the specimens (e.g., both the largest and the smallest measured *Proterochersis robusta*, SMNS 16442 and SMNS 16603, respectively, are among the specimens with the largest difference between the two size estimation methods, while the smallest complete *Proterochersis porebensis*, ZPAL V. 39/34, is the only specimen with a complete agreement between them). Likewise, although some inaccuracy due to not accounting for the complete shell geometry is undeniable, both extreme values of the ratio between the estimations base on the two methods are represented by fragmentary specimens (SMNS 11396 and SMNS 17755) while the most complete shell (SMNS 17561) shows a nearly perfect average value for the species ($0.87786 \approx 0.88$). Therefore, a simple intraspecific variability of the relative breadth of the femoral scute appears to be the most convincing explanation. Curiously, *Proterochersis porebensis* presents, on average, more consistency between both estimation methods (ratio between the results above 0.9 for all three complete specimens, average ratio = 0.96, average ratio when the juvenile ZPAL V. 39/34 is excluded = 0.89).

## Discussion

### Taxonomic affinity of SMNS 15479

Out of the eight named taxa of Triassic turtles, seven preserve plastra. SMNS 15479 cannot be directly compared morphologically with *Chinlechelys tenertesta* Joyce et al. 2009 [67] but is significantly larger than all specimens of that turtle known thus far and the record of *Chinlechelys tenertesta* is currently restricted to the Bull Canyon Formation of the USA [66–68]. The australochelyids, *Palaeochersis talampayensis* and *Waluchelys cavitesta*, in the Triassic were not noted outside of the Los Colorados and Quebrada del Barro formations of Argentina, respectively [31–34, 78]. Whereas the scute sulci pattern in the australochelyids is typically very poorly preserved and mostly not traceable, the representatives of that clade lack the caudal processes of the plastron and their posterior plastral lobes are distinctly tapering posteriorly and narrower than the posterior plastral lobe of SMNS 15479. '*Proganochelys*' *ruchae* is thus only known from the Huai Hin Lat Formation of Thailand and although the comparison is difficult due to the fragmentary nature of the available material, the distinct lateral extragular projections seem to be absent in SMNS 15479 [20, 70]. The same character seems to distinguish SMNS 15479 from the proganochelyid from the Fleming Fjord Formation of Greenland [39, 41]. Only four Triassic turtle species identified thus far lived in Central Europe during Norian: *Proganochelys quenstedtii*, *Keuperotesta limendorsa*, *Proterochersis robusta*, and *Proterochersis porebensis*. The former three have a confirmed record in the Löwenstein Formation of Germany, precisely in the proximity of Stuttgart [11, 12, 17, 18, 46, 48, 50, 54, 55] while the latter

comes from the Grabowa Formation of Poland [14, 63, 64]. Based on the morphology of the plastron, *Proganochelys quenstedtii* can be easily excluded based on the absence of the second pair of abdominal, intercaudal, and caudal scutes, as well as based on a divergent morphology of the anterior plastral lobe and relative positions and shape of scute sulci [18, 36, 37]. *Keuperotesta limendorsa* differs from SMNS 15479 in a more posterior position of the humeropectoral sulcus [14] and broader anterior plastral lobe with nearly parallel lateral edges. Other than that, the plastral morphology of that species is very similar to that of *Proterochersis* spp. The morphology of the plastra is pretty variable in *Proterochersis* spp. [15] and thus far *Keuperotesta limendorsa* is only represented by a single specimen (SMNS 17757) so its intraspecific variability is not clear, but at the moment in terms of the morphology of its anterior plastral lobe it falls outside of the spectrum observed in *Proterochersis* spp. and SMNS 15479. Therefore, SMNS 15479 can be safely referred to the genus *Proterochersis*. A precise specific attribution is, however, more problematic. Despite small but consistent differences in the carapace [14–16], no morphological characters in the plastron were identified that could allow distinguishing both species. Size-wise, SMNS 15479 fits within the range documented in *Proterochersis porebensis* and exceeds the sizes of all *Proterochersis robusta* specimens found thus far [15]. The average femoral scute width to plastral length ratio in *Proterochersis porebensis* is closer to the ratio observed in SMNS 15479, especially in the juvenile ZPAL V. 39/34. Given that no specimens morphologically attributable to *Proterochersis porebensis* were found in Germany despite numerous specimens discovered during over 160 years of exploration [48, 54] and that SMNS 15479 comes from the region and formation which yielded multiple specimens of *Proterochersis robusta* (but also the locality which produced no other turtle specimens), we tentatively identify it as an exceptionally large and presumably ontogenetically old specimen of the later species, noting however, that morphologically it is not diagnostic beyond *Proterochersis* sp.

## Shell pitting

The anteromedial part of the anal scute areas of SMNS 15479 is abnormally pitted (Figs 7A–7D, 9). In the absence of actual bone material, interpretation of these marks is difficult, and should be considered tentative. However, if the natural mold preserves their morphology accurately (as is assumed for the non-pathological regions of the same specimen and as confirmed for those regions based on comparisons with other specimens), they can provide some useful information. The limited, cluster-like distribution of the pits, and their rarity (apparent absence in other fossils from the Löwenstein Formation) seem to preclude their identification as erosion traces left postmortem by some common biotic or abiotic factor. Predator or scavenger bite marks may produce similar morphologies [82, 83] and large carnivores, such as phytosaurs and pseudosuchians, have been found in the Löwenstein Formation [55, 84–88], the location of the pits close to the middle of the plastron of this exceptionally large individual, their layout, smooth edges, and lack of scratches, brittle deformation, or evidence of gnawing seem to reject their identity as bite marks. They are, however, morphologically consistent with *Karethraichnus lakkos* Zonneveld et al. 2015, a relatively common ichnospecies of turtle ectoparasite-caused bone modifications [82, 89–91]. Although mostly associated with aquatic ectoparasites, such as leeches or barnacles [82, 89–91], traces morphologically corresponding to *Karethraichnus* ispp. can be also caused in the terrestrial environment by ticks [82, 89, 90, 92–94]. Although *Proterochersis robusta* is mostly considered a terrestrial turtle [13, 56, 95, 96], an aquatic or semiaquatic ecology was also proposed for that species [96, 97] and *Proterochersis porebensis* seems to have preferred at least a partially aquatic habitat [52, 53, 57]. The location of the traces in the middle part of the plastron would, nonetheless, protect any potential pitting

agents from the heat and increased drying caused by direct sunlight. If identification of the pits as shell pathologies is correct, then SMNS 15479 would not only constitute a very rare case of a preservation of a turtle plastron as a natural external mold, but also a negative of a named ichnospecies and, in addition, extend its known record from beyond the Cenozoic, into the Late Triassic. Nonetheless, as explained above, due to the unusual mode of preservation and lack of actual bone, we exercise caution and mark this notion as hypothetical. Although rare, potentially similar pit-like traces are also found on the external (scute-covered) turtle shell fragments from the Grabowa Formation of Poland, but this material awaits description.

## Taphonomy of natural molds of bones

The phenomenon of preservation of natural molds or impressions of bones in the fossil record is not exceptionally rare [12, 13, 17, 98–109]. Nonetheless, such specimens constitute a minor percentage of described fossils, and their formation is poorly studied. The occurrence of the natural bone molds appears to be heavily locality- or formation-specific and possibly involves particular conditions (grain size and composition of the sediment, mineral content of the permeating solution, etc.), but these conditions are poorly documented and understood.

Several general types of natural molds can be distinguished. The first category are molds formed in situ and prior to exposure, after the dissolution of the bones still embedded in the matrix due to the diagenetic setting which is unfavorable for bone preservation [99, 102]. In some cases, articulated or semiarticulated skeletons can be preserved that way and the entire specimen can be split into two slabs capturing all its surfaces [100, 104, 105, 108–110]. The second category are cases in which the bone is initially preserved (i.e., the environment is favorable to bone preservation) but either subsequently destroyed by natural processes such as weathering, separated during recovery or preparation, or destroyed due to improper handling. In those cases, the preserved bone must be exposed to the environment, therefore usually the morphology of only one side or surface is preserved, the other being destroyed. This category includes, e.g., *Proganochelys quenstedtii* GPIT-PV-30000 and *Proterochersis robusta* SMNS 11396 and SMNS 16442 [12, 13, 17–19]. In rare cases, more complex scenarios can occur, for example formation of a natural cast of the fossil by filling the natural mold with new sediment [99, 101, 102, 106], or selective preservation of some bones but dissolution of others [100].

Probably the most common finds of that kind are complete or partial internal molds of the shells of turtles [10, 11–18, 22, 111–116]. This unique predisposition for the formation of natural molds of turtle shells, especially of the interior (steinkerns), was not explored in detail, but several influencing factors may be proposed. Firstly, the interior of the shell may form a partially enclosed space separated from the surrounding environment. As a result, it may (1) promote the entrance of finer, easier to transport sediment over more coarse material before the burial, acting as a form of sorting; (2) trapping the infilling material even during limited transport, possibly further differentiating the infilling and surrounding sediment; and (3) impacting the chemistry of the sediment during deposition and fossilization. Note, however, that natural internal molds of carapaces occur also in turtles lacking an osseous bridge between the carapace and the plastron [115], and therefore have no significant barriers for the exchange of the sediment on both sides of the carapace. Moreover, frequently, at least in the case of Triassic taxa from the Löwenstein Formation, there is no macroscopically noticeable difference between in lithology of the rock matrix inside and outside of the shell. Secondly, the relative smoothness and lack of protruding elements in the turtle shells may provide natural separation planes, facilitating their detachment from large, plain surfaces of the underlying (or overlying) rock matrix. The relative frequency of the steinkerns over the external shell molds may be caused by the usually much larger smoothness of the visceral surfaces of the shell bones,

compared to their external surfaces, and by the common presence of lamellar or highly organized parallel-fibered bone tissue on their visceral surfaces [56, 57, 117], possibly allowing easier delamination and separation of the elements along those surfaces. A collection bias may also play a part, especially in cases when the potential external mold may be significantly bulkier, heavier, more fragile and/or more fragmented than the internal mold. Finally, due to the usual relatively even, low thickness, exposed shells are mostly superficial relative to the enclosed matrix and may be more uniformly impacted by weathering or other destructive environmental factors, much easier than in the case of typical skeletal specimens, being usually more complex and irregular and extending deeper into the rock. In contrast to invertebrate internal molds [118–121], in most cases the turtle steinkerns appear to result from physical (mechanical) damage to or removal of the bone rather than its chemical dissolution, as evidenced by the frequent retention of minute bone remains with jagged edges or only partial exposure of the steinkern from the encapsulating articulated shell. The compact, rounded form of the steinkerns, with a relatively low surface to volume ratio, in connection with their sandstone-based matrix may be more resistant to weathering. Analogously, large, isolated, sandy natural footprint casts of ichnotaxa, such as the Permian *Pachypes dolomiticus*, which resist weathering, are found more commonly than natural footprint molds on finer sediment [122, 123]. Also, a similar preservation is observed in chirotheriid tracks, such as the Triassic *Brachychirotherium hassfurtense* Beurlen, 1950 [124–127].

Among the most famous formations with a good record of natural molds of terrestrial tetrapod remains are the Permian Cutties Hillock Formation and the Late Triassic Lossiemouth Sandstone Formation from Scotland [98, 103, 104, 107, 109, 128]. The Cutties Hillock Sandstone Formation consists of two units: pebbly (sheet flood deposits) and cross-bedded coarse sandstones (fossil barchan dunes) with less numerous quartz pebbles [104]. As noted by Newton [109], the sediment lining the molds is often darkened and covered with a layer containing iron, manganese, and cobalt, and in some cases a secondary infilling of iron oxide may be formed. The molds usually represent well preserved and articulated skeletons or their fragments, although some exhibit compression or pre-burial damage [104, 109]. The yellowish or pinkish, uniform-grained sandstones of the Lossiemouth Sandstone Formation are finer than those of the Cutties Hillock Sandstone Formation and generally lack pebbles [104, 109]. Some large specimens from the Lossiemouth Sandstone Formation exhibit signs of transport or deformation, in some cases affecting only some bones in the association [98, 104]. Bones, if present, are soft and may be partly replaced by goethite and fluorite–this form of preservation is locality-dependent and appears to be associated with a lesser hardness of the sandstone [98, 104].

In some regards (the presence of pebbles, discoloration of the sediment surrounding the cavities of the molds), the Cutties Hillock Sandstone Formation appears to represent a more similar taphonomic and diagenetic setting to the Löwenstein Formation than the Lossiemouth Sandstone Formation. However, bone preservation in the Löwenstein Formation is usually good and the bones do not show any evidence of dissolution prior to their exposition to external factors. Nonetheless, in all three formations, the dominating type of fossil-containing rock matrix is sandstone, which may suggest that matrix of that particular type is preferable for the retention of the shape of embedded fossils, regardless of other mechanisms of bone preservation or destruction. However, sandstone matrix (at least in some cases yielding animal body fossils) is common in nearly all Triassic turtle-bearing formations, most of which show no potential for production of natural molds (Table 1), so other factors must also be at play. Interbedding with finer-grained rocks, such as mudstones, claystones, marlstones, or limestones, may also be beneficial for formation of natural molds. In the case of footprints, the interplay between sandstone and very fine-grained sediment may significantly impact the preservation.

In such cases, the sandstone provides support and the fine sediment captures finer details [129, 130]. Possibly, the process may be analogous in the case of the turtle steinkerns and external molds from the Triassic of Germany.

## Acknowledgments

We thank Gabriela Cisterna (PULR), Laura Cotton (NHMD), Zheng Fang (IVPP), Eudald Mujal Grané (SMNS), Chun Li (IVPP), Ning Li (IVPP), Bent Erik Kramer Lindow (NHMD), Jun Liu (IVPP), Rainer Schoch (SMNS), Daniela Schwarz (MB), Rolf Schweizer (CSMM), and Heike Straebelow (MB) for the access to the specimens housed at their respective institutions, as well as Nils Natorp (Geocenter Møns Klint) and Volker Neipp (Museum Auberlehaus in Trossingen) for the access to the Greenland proganochelyid NHMD 190349 and *Proganochelys quenstedtii* SMNS 17204 on their expositions. Tomasz Sulej (ZPAL) and all the participants in the excavations in Poręba are thanked for their contribution in the collection and preparation of the material of *Proterochersis porebensis*. No permits were required for the described study, which complied with all relevant regulations.

## Author Contributions

**Conceptualization:** Tomasz Szczygielski.

**Data curation:** Tomasz Szczygielski.

**Formal analysis:** Tomasz Szczygielski, Lorenzo Marchetti.

**Funding acquisition:** Tomasz Szczygielski.

**Investigation:** Tomasz Szczygielski, Lorenzo Marchetti.

**Methodology:** Tomasz Szczygielski.

**Project administration:** Tomasz Szczygielski.

**Supervision:** Tomasz Szczygielski.

**Validation:** Tomasz Szczygielski.

**Visualization:** Tomasz Szczygielski, Dawid Dróżdż.

**Writing – original draft:** Tomasz Szczygielski.

**Writing – review & editing:** Tomasz Szczygielski, Lorenzo Marchetti, Dawid Dróżdż.

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
