## [Decision Letter · Decision Letter 0]

11 Jan 2024

PONE-D-23-40099Natural external plastron mold of the Triassic turtle Proterochersis: an unusual mode of preservationPLOS ONE

Dear Dr. Szczygielski,

Thank you for submitting your manuscript to PLOS ONE. After careful consideration, we feel that it has merit but does not fully meet PLOS ONE’s publication criteria as it currently stands. Therefore, we invite you to submit a revised version of the manuscript that addresses the points raised during the review process.

We look forward to receiving your revised manuscript.

Kind regards,

Dawid Surmik, PhD

Academic Editor

PLOS ONE

Journal Requirements:

3. In your manuscript, please provide additional information regarding the specimens used in your study. Ensure that you have reported human remain specimen numbers and complete repository information, including museum name and geographic location. 

For more information on PLOS ONE's requirements for paleontology and archeology research, see https://journals.plos.org/plosone/s/submission-guidelines#loc-paleontology-and-archaeology-research.

4. Thank you for stating the following financial disclosure: "The study was supported by the National Science Centre, Poland (Narodowe Centrum Nauki, https://www.ncn.gov.pl/en) grant no. 2020/39/B/NZ8/01074 awarded to T. Sz."

Reviewers' comments:

Reviewer's Responses to Questions

**Comments to the Author**

1. Is the manuscript technically sound, and do the data support the conclusions?

Reviewer #1: Partly

Reviewer #2: Yes

2. Has the statistical analysis been performed appropriately and rigorously? 

Reviewer #1: N/A

Reviewer #2: N/A

3. Have the authors made all data underlying the findings in their manuscript fully available?

Reviewer #1: Yes

Reviewer #2: Yes

4. Is the manuscript presented in an intelligible fashion and written in standard English?

Reviewer #1: Yes

Reviewer #2: Yes

5. Review Comments to the Author

Reviewer #1: Dear Authors. Find attached my MODERATE comments and suggestions to your manuscript. I hope you find them usefull and in order to improve your manuscript. My major concerns is the interpretation of the bioerosional marks, as you don't have bone preserve, this should be consider as very hypothetical.

Sincerely,

Reviewer #2: Dear colleagues,

I have carefully read the manuscript by Szczygielski and collaborators, which deals with peculiar turtle shell fossils in terms of preservation and taphonomic pathways.

I find the descriptions of the fossils accurate and the interpretations thoroughly discussed, providing a compelling investigation on the taphonomy of steinkerns. Figures are nicely presented and appropriate.

I only found minor corrections to be made. I highlighted them in the attached PDF by using the Adobe Reader Comment Tools. As a suggestion, the authors can add chirotheriid footprints as a similar case of preservation (besides Pachypes, which is already very interesting). I find this comparison interesting, expanding also the research to ichnology. The identifiaction of potential pathologies in this type of preservation is also relevant; the authors provide a different perspective when examining this type of fossils that allows identifying more features on their paleobiology.

All in all, I think this is a very well executed research work, and I hope to seeing it published after a minor revision is undertaken.

Best regards,

Eudald Mujal (Staatliches Museum für Naturkunde Stuttgart)

6. PLOS authors have the option to publish the peer review history of their article (what does this mean?). If published, this will include your full peer review and any attached files.

Reviewer #1: **Yes: **Edwin Cadena

Reviewer #2: **Yes: **Eudald Mujal

---

## [Author Response · Author response to Decision Letter 0]

27 Jan 2024

Dear Editor, dear Reviewers,

Thank you for your helpful comments. We addressed them by incorporating all minor corrections verbatim into the text using the track changes option and responding to other points in more detail below (Reviewers’ comments are cited in bold, our responses in regular typeface). The figures are amended as suggested by the Reviewer 2, and references to the new Fig 9 were adder in relevant places of the manuscript.

Yours sincerely,

Tomasz Szczygielski

REVIEWER 1 (Edwin Cadena)

Dear Authors. Find attached my MODERATE comments and suggestions to your manuscript. I hope you find them usefull and in order to improve your manuscript. My major concerns is the interpretation of the bioerosional marks, as you don't have bone preserve, this should be consider as very hypothetical.

Sincerely,

Thank you very much for the comments! The uncertainty regarding the interpretation of those marks is now highlighted in a clearer way and the discussion was supplemented, following suggestions. We also modified Fig 8 and added new Fig 9, as requested. Please see detailed responses below.

Page 25, lines 351–353: “No, the scutes are not preserved. If you say they are preserved, it means preservation of their original keratins constituent are preserved, or something tranformed during fossildiagenesis with some degree of thickness. What is preserved is their silhouette or outlines.”

That is of course true, thank you, we corrected the wording in that part.

Page 25, lines 373–374: “Alternatively, this could be just a matter of differential compactness of the sediment/rock matrix during fossildiagenesis, especially at the anterior lobe region.”

Potentially – yes, of course. But (1) the general shape observed in SMNS 15479 is consistent with that of other specimens of Triassic turtles, including those from other formations, and (2) turtle shells and steinkerns from the Löwenstein Formation typically present no significant deformations. Therefore, we consider the preserved geometry to be at least predominantly original and unaffected. We added a sentence explaining this.

Page 26, lines 378–379: “another indication that the dept is product of diganesis. 2 cm difference between the gulars-extragulars and the pectorals level.”

Actually no, this is normal and expected. The difference in the alignment and depth of the gulars + extragulars and humerals due to the S- or Z-shaped anteriormost end of the plastron is observed in all proterochersids and in Proganochelys quenstedtii, regardless of geological formation, no deformation needed.

Page 33, line 541: “or during the burial of the specimen. The question here is, do you believe the keratinous layers that formed the scutes was pealed off from the bone and buried? or the specimen was buried complete bone/scutes and then during fossildiagenesis pre/post the bone was eroded?. This could have alterated the symetry of the parts of the imprinted layer of the scutes.”

The most likely scenario is that the specimen was either buried complete with the scutes still attached, or the bony plastron was buried after the scutes detached, but in any case we consider the preserved impression to most likely represent the bone surface rather than scute surface – this is suggested by the appearance of breaks and cracks crossing the sulci and continuing across more than one scute area. In both cases the bone would be removed last but the impact on the relative proportions between the width of the plastron at the level of the femoral scutes and the plastron length should be deniable. Note that in proterochersids the plastron in that region is virtually flat (no medial concavity etc.) so no reasonable amount of flattening due to compaction should noticeably increase its width. If anything, flattening could increase the width at the level of abdominals and should extend the length of the plastron due to redistribution of bones involved in the formation of the central concavity and the S-shaped anterior lobe, but this is not observed in that case.

Page 37, lines 624–627: “How can you be 100% this bioerosional attacks ocurred in live?. Carcases of turtles could move to water bodies by currents and expose to additional bioerosion.”

We added a relevant explanation. Of course – we cannot be 100% sure (and most likely we would not be if we had the actual bones preserved, either, because the origin of some traces is difficult to establish based on external examination, and the size and shape of the specimen would not allow performing tomography or histological thin sections). But we added a short explanation why we consider the postmortem origin of those pits less likely. Firstly, the pits are present on a relatively small surface, packed in a cluster-like arrangement, and there are no similar marks on other fossils from the same formation that we know of. This seems to refute the idea that they were caused by some common biotic or abiotic factors – in such a case, similar pits should be more common in the same depositional setting, and the distribution should be more uniform across the specimen. Secondly, there are similar unpublished traces known from the turtle shell fragments from Grabowa Formation, and they are restricted to external (scute-covered) surfaces, which suggests that only those surfaces were accessible at the time of their formation – that seems to support the in vivo origin of the traces. And finally, the morphology, distribution, and size of the pits is consistent with Karethraichus lakkos – this was already stated in the initial submission. We are not aware of post-mortem factors that would produce similar, smooth-edged, round pits in dead bones in a freshwater environment.

Page 38, lines 631–632: “this is very speculative. It could be also due to preburial degradetation of the keratinous layer of the scutes and not necessarly biogenic in origin. Degradation of the thin keratinous layer is very typical in turtles, and it can stats as circular spots. per. orbservation. Bioerosional marks can be 100% establish only when bone is preserved or original cuticular layer for example in fossil leaves.”

Please note that we do not postulate that the keratinous layer was at all present in the specimen at the moment of burial, and even if it was, the presumed pits were deeper then surrounding sulci, so deep enough to reach the underlying bone. Therefore, scute degradation alone is not a sufficient explanation. However, we modified the respective sections to stress the tentativeness of our interpretation.

Page 39, lines 659–660: “this could have be tested for your specimen, doing a thin sections of the external rock matrix and internal mold rock matrix.

This is a good idea, however, please note that our specimen represents an external mold, and the commented sentence refers to internal molds. As mentioned below, we did not attempt destructive sampling at this time.

Page 39, line 667: “how this was tested? thin sections and observation under the microscope?. references if possible.”

No, we did not perform destructive sampling. For clarity, we specified that there is no macroscopically noticeable difference in lithology.

Caption to Fig 1: „dorsal”

In fact, the presented surface was in life aligned anterodorsally rather than dorsally, but we prefer the term ‘external’ here, because in high-domed shells of turtles such as Proterochersis robusta the direction of the scute-covered (external) surfaces is highly variable across the carapace (e.g., the mesial part of the third vertebral was directed dorsally but the lateral edges of the same scute were directed dorsolaterally, some parts of the pleurals were directed either dorsolaterally, anterodorsolaterally, posterodorsolaterally, or completely laterally, the fifth vertebral was directed nearly completely posteriorly in the posterior part but posterodorsally in the anterior part, etc. In the case of isolated, sometimes crushed or deformed shell fragments, the original inclination is sometimes difficult to establish and to reproduce in figures. For that reason, we consider the word ‘external’ to be more intuitive and to evoke the intended meaning better than the word ‘anterodorsal’.

Fig 7: “what is your interpretation of this line? if it a fracture, that could be adding longer measurements of the specimen.”

Yes, we consider this a fracture. Comparison with other, unbroken specimens suggests that its impact on the measurements of the specimen is not significant. But please note that the specimen is wider than expected, not longer than expected, therefore even if its length was extended because of this break, that would mean that the width to length ratio was initially even larger than discussed.

Fig 7: “is it also a spot for "bioerosion" or keratinous layer degradation?”, “Another spot?”

Please note that the specimen in this figure is shown as a digital negative, i.e., convexities of the actual slab show here as concave and concavities – as convex structures (as they would in the original, imprinted plastron). The spots indicated were either linear concavities (and thus most probably breaks) or convex (possibly clasts between the imprinted surfaces of the plastron and the surface of sediment). The traces we described are concave in this figure (therefore we describe them as pits). As such, they do not present the same type of morphology (we imagine this may be difficult to see in the review PDF but should be clear in the final image at 600 DPI).

Fig 7: “A close-up of this region will be very important to show. Is it margin straight? or it has some level of difuminating thicknes?.”

We are not certain what exactly the Reviewer meant here, so we added a new figure presenting a closeup of most of the plastron imprint surface on the slab, and a closeup of the plaster cast of the pitted area (Fig 9). The margin of the imprint is jagged and the imprint surface falls into rough, mostly recessed surface of eroded sandstone. This appears to be an effect of damage which occurred after the imprint was exposed. The edges of the pits, on the other hand, are rounded and smooth.

Fig 8: “Some labels are hard to see for example here I, C, An maybe try to add a black shadow or use a line”, “hard to see E, G,”

Black outlines were added for improved readability.

REVIEWER 2 (Eudald Mujal)

Dear colleagues,

I have carefully read the manuscript by Szczygielski and collaborators, which deals with peculiar turtle shell fossils in terms of preservation and taphonomic pathways.

I find the descriptions of the fossils accurate and the interpretations thoroughly discussed, providing a compelling investigation on the taphonomy of steinkerns. Figures are nicely presented and appropriate.

Thank you very much! We are happy to see that, especially that the Reviewer is familiar with the described material, which is housed in his employing institution.

I only found minor corrections to be made. I highlighted them in the attached PDF by using the Adobe Reader Comment Tools. As a suggestion, the authors can add chirotheriid footprints as a similar case of preservation (besides Pachypes, which is already very interesting). I find this comparison interesting, expanding also the research to ichnology. The identifiaction of potential pathologies in this type of preservation is also relevant; the authors provide a different perspective when examining this type of fossils that allows identifying more features on their paleobiology.

The chirotheriids were added to the discussion, as suggested. All the suggestions from the PDF were introduced to the text.

All in all, I think this is a very well executed research work, and I hope to seeing it published after a minor revision is undertaken.

Best regards,

Thank you! 

Page 39, line 671: “Could it also be a matter of exposition? The external shell is more easily exposed than the internal area, so that it can be weathered/destroyed more easily. I understand this is what you suggest some lines below (from line 675-676 on), right?”

Yes, exactly, this is what we meant there.

---

## [Decision Letter · Decision Letter 1]

8 Feb 2024

Natural external plastron mold of the Triassic turtle Proterochersis: an unusual mode of preservation

PONE-D-23-40099R1

Dear Dr. Szczygielski,

We’re pleased to inform you that your manuscript has been judged scientifically suitable for publication and will be formally accepted for publication once it meets all outstanding technical requirements.

Kind regards,

Dawid Surmik, PhD

Academic Editor

PLOS ONE

Additional Editor Comments (optional):

Reviewers' comments:

Reviewer's Responses to Questions

**Comments to the Author**

1. If the authors have adequately addressed your comments raised in a previous round of review and you feel that this manuscript is now acceptable for publication, you may indicate that here to bypass the “Comments to the Author” section, enter your conflict of interest statement in the “Confidential to Editor” section, and submit your "Accept" recommendation.

Reviewer #1: All comments have been addressed

Reviewer #2: All comments have been addressed

2. Is the manuscript technically sound, and do the data support the conclusions?

Reviewer #1: Yes

Reviewer #2: Yes

3. Has the statistical analysis been performed appropriately and rigorously? 

Reviewer #1: N/A

Reviewer #2: N/A

4. Have the authors made all data underlying the findings in their manuscript fully available?

Reviewer #1: Yes

Reviewer #2: Yes

5. Is the manuscript presented in an intelligible fashion and written in standard English?

Reviewer #1: Yes

Reviewer #2: Yes

6. Review Comments to the Author

Reviewer #1: Dear Authors.

I am glad you found most of the comments and suggestions useful and incorporated them into the new version of your manuscript.

Sincerely yours,

Reviewer #2: The revised version of the manuscript addresses, in my opinion satisfactorily, all the comments of both reviewers. In this sense, I do not think further modifications are necessary. I thank the authors for their efforts to follow all the suggestions, and congratulate them for their work.

All the best,

Eudald Mujal (Staatliches Museum für Naturkunde Stuttgart)

7. PLOS authors have the option to publish the peer review history of their article (what does this mean?). If published, this will include your full peer review and any attached files.

Reviewer #1: **Yes: **Edwin Cadena

Reviewer #2: **Yes: **Eudald Mujal

---

## [Editor Report · Acceptance letter]

20 Mar 2024

PONE-D-23-40099R1 

PLOS ONE

Dear Dr. Szczygielski, 

I'm pleased to inform you that your manuscript has been deemed suitable for publication in PLOS ONE. Congratulations! Your manuscript is now being handed over to our production team.

Kind regards, 

on behalf of

Dr. Dawid Surmik 

Academic Editor

PLOS ONE